# Improvement of cryo-EM maps by simultaneous local and non-local deep learning

Jiahua He [1], Tao Li[1] & Sheng-You Huang [1] ✉

Cryo-EM has emerged as the most important technique for structure determination of macromolecular complexes. However, raw cryo-EM maps often exhibit loss of contrast at high resolution and heterogeneity over the entire map. As such, various post-processing methods have been proposed to improve cryo-EM maps. Nevertheless, it is still challenging to improve both the quality and interpretability of EM maps. Addressing the challenge, we present a three-dimensional Swin-Conv-UNet-based deep learning framework to improve cryo-EM maps, named EMReady, by not only implementing both local and non-local modeling modules in a multiscale UNet architecture but also simultaneously minimizing the local smooth L1 distance and maximizing the non-local structural similarity between processed experimental and simulated target maps in the loss function. EMReady was extensively evaluated on diverse test sets of 110 primary cryo-EM maps and 25 pairs of half-maps at 3.0–6.0 Å resolutions, and compared with five state-of-the-art map post-processing methods. It is shown that EMReady can not only robustly enhance the quality of cryo-EM maps in terms of map-model correlations, but also improve the interpretability of the maps in automatic de novo model building.

Cryogenic electron microscopy (cryo-EM) has become one of the standard techniques for structure determination of large biological complexes in structural biology[1–5], owing to its advances in hardware[6] and image processing algorithms[7–15]. The goal of cryo-EM is to obtain the atomic models of macromolecular complexes from the EM density maps, where the quality of EM maps is critical[16–27]. However, due to some inherent impacts from sources like molecular motions, heterogeneity, and imperfect imaging, raw cryo-EM maps often face loss of contrast at high-resolution and would not be immediately ready for accurate structure determination. To address this problem, various approaches have been developed to improve the map quality by sharpening or modifying the density during the post-processing step of cryo-EM workflow[28–36].

Traditional approaches for map sharpening can be roughly grouped into two categories, global approaches and local approaches. Global sharpening approaches normally determine a single B-factor across an EM map and apply the same density correction to the whole map. phenix.auto_sharpen[29] is such a global sharpening method[26] based on the optimization of details and connectivity of sharpened maps. Similar B-factor correction algorithm is also adopted by RELION post-processing[30]. Local sharpening approaches take heterogeneity in cryo-EM maps into account and adopt local density-dependent correction during map sharpening. LocScale[31] is a general procedure for local sharpening of cryo-EM density maps based on prior knowledge of an atomic reference structure. LocalDeblur algorithm[32] employs a Wiener restoration approach that performs local deblurring with a strength proportional to an estimation of the local resolution[37]. Density modification[33,34] maximizes a combined likelihood function that incorporates both the plausibility of the map and the agreement with experimental values. LocSpiral[35] employs the spiral phase transformation to factorize the volume and locally enhances high-resolution features of cryo-EM maps. All these sharpening algorithms have

[1]School of Physics and Key Laboratory of Molecular Biophysics of MOE, Huazhong University of Science and Technology, Wuhan, China.
✉e-mail: huangsy@hust.edu.cn

achieved great successes in post-processing cryo-EM maps. However, there exist drawbacks in current approaches. Namely, global sharpening approaches may result in over-sharpened and under-sharpened regions because of different signal to noise ratios in EM maps, and some local sharpening approaches rely on the use of a priori information about the target map like masks to distinguish the macromolecule from the noise, estimation of the local resolution of the map, and/or structural information of atomic models.

Addressing the limitations in traditional sharpening approaches, deep learning-based methods have recently been proposed for automatic cryo-EM volume post-processing to improve the interpretability of cryo-EM maps. Deep cryo-EM Map Enhancer (DeepEMhancer)[38], which is a fully automatic deep learning-based methods, mimics the local sharpening effect of the LocScale algorithm. EM-GAN[39] is another deep learning-based method that uses a three-dimensional generative adversarial network (GAN) for generating an improved-resolution EM map from an experimental EM map. In spite of their good performances on some experimental cryo-EM maps, both DeepEMhancer and EM-GAN face their respective challenges. DeepEMhancer might be limited by the accuracy or noise of experimental cryo-EM maps in the training set as it uses atomic model-guided optimized experimental maps as the target maps during the training. EM-GAN tries to minimize the mean difference between the voxels from the generator network and those in the simulated maps, which may miss the structural correlation between two groups of voxels in terms of density contrast.

To overcome the shortcomings of existing approaches, we here present a three-dimensional deep learning framework to improve the interpretability of cryo-EM maps by simultaneous local and non-local deep learning, which is referred to as EMReady, aiming to make the EM map ready for atomic structure determination. EMReady adopts the three-dimensional (3D) Swin-Conv-UNet-based network architecture (SCUNet)[40], which combines the advantages of conventional residual convolution for local modeling, swin (shifted window) transformer for non-local modeling, and multiscale UNet for further enhancement of local and non-local modeling. The swin transformer[41] is an efficient transformer that combines self-attention of non-overlapping local windows and non-local cross-window connection by shifted window partitioning. With the SCUNet architecture, EMReady is capable of capturing the non-local features within each input density slice of size 48 Å × 48 Å × 48 Å. The local and non-local modeling of EMReady is implemented not only in the network architecture but also in the training process. Specifically, our network is trained by simultaneously minimizing the local smooth L1 distance and maximizing the non-local structural similarity (SSIM) between processed experimental and simulated target maps. Compared with the simple smooth L1 loss, incorporating the SSIM loss in the training process can effectively prevent the network from possible overfitting. During the training of EMReady, we use the simulated cryo-EM maps from their associated structures from the Protein Data Bank (PDB)[4] instead of LocScale-processed experimental cryo-EM maps as the target maps, which can avoid the impact of noise in the experimental maps. EMReady is extensively evaluated on diverse test sets of primary EM maps and half-maps. It is shown that EMReady is able to not only robustly enhance the quality of cryo-EM maps in terms of various map quality metrics[42], but also lead to better structure models built by phenix.map_to_model[17–19] and MAINMAST[20], demonstrating the improvement of EMReady in both the quality and interpretability of cryo-EM maps.

## Results
### Overview of EMReady
Figure 1 shows an overview of the EMReady workflow. The SCUNet adopted by EMReady consists of three encoder, one bottleneck, and three decoder swin-conv (SC) blocks with skip connections. A non-redundant set of 460 experimental cryo-EM maps is obtained from the

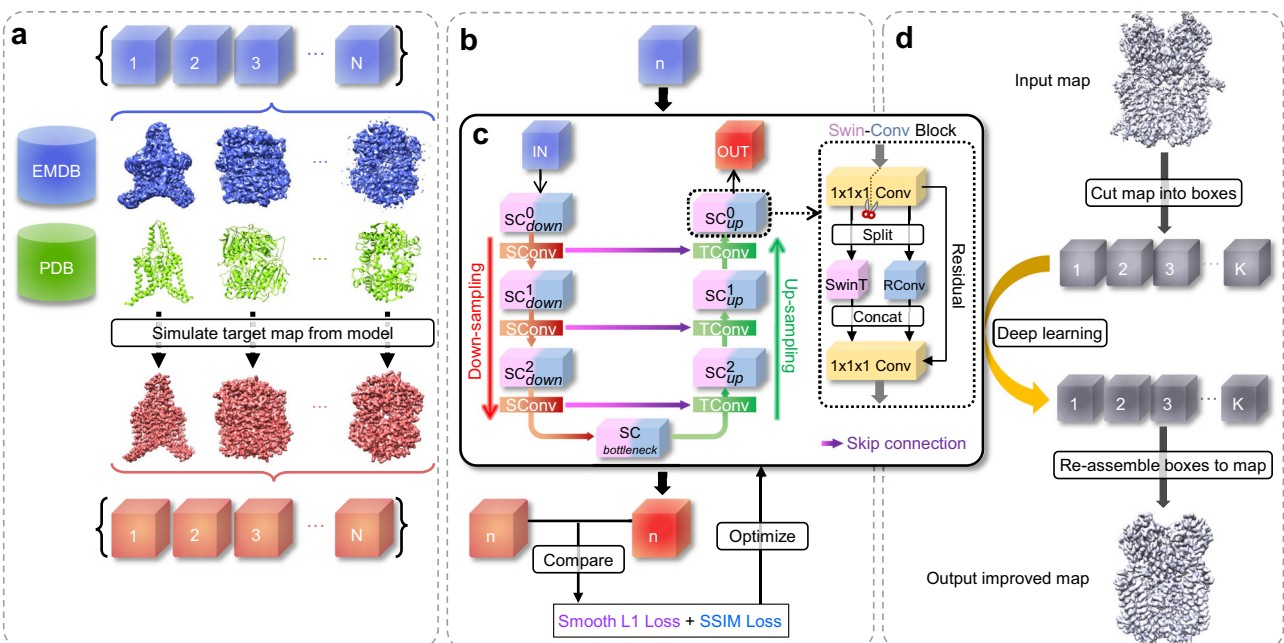

**Fig. 1 | Overview of the EMReady deep learning framework. a** Preparation of the training data. EM density maps and their associated PDB models are downloaded from the EMDB and PDB, respectively. Target maps are simulated from the PDB models. The experimental maps and simulated maps are then cut into pairs of experimental boxes and simulated boxes. **b** The training procedure of EMReady. In each training round, an experimental box is input to the deep learning model, and the processed box is compared with its corresponding simulated box. A combination of local smooth L1 loss and non-local SSIM loss is used to optimize the deep learning model through backpropagations. **c** The schematic of the SCUNet architecture used in EMReady. A given input EM density box will go through a UNet-like encoder-decoder network, where swin-conv (SC) blocks are used as the main building block. Swin transformer (SwinT) for non-local modeling and residual convolution (RConv) for local modeling are implemented in parallel in each SC block. **d** The map processing workflow of EMReady. For a given input EM density map, EMReady first cuts it into boxes. All the boxes are processed by the trained deep learning model in (**c**), and then re-assembled to the output processed map.

Electron Microscopy Data Bank (EMDB)[5], of which 280 maps are used as the training set, 70 maps are used as the validation set, and the other 110 maps are used for testing. The corresponding atomic structures of the 350 EM maps are taken from the Protein Data Bank (PDB)[4]. The grid size of the maps is unified to 1.0 Å by a cubic interpolation. For training, experimental maps and simulated maps from the corresponding PDB structures are cut into pairs of experimental volume slices and simulated slices of size $48 \times 48 \times 48$. During the training, an experimental volume slice is processed by the network and the comparison between the processed slice and the simulated slice is carried out. The differences between processed slices and simulated slices measured by a combined loss function of smooth L1 loss and SSIM loss are used to iteratively optimize the network through backpropagations. During the evaluation, a given cryo-EM density map is cut into small slices and each slice is processed by EMReady. Then, the output processed slices are re-assembled into the final density map.

### Table 1 | Map quality on the test set of 110 deposited primary maps

| Method | FSC-0.5 (Å) | Q-score | CC_box | CC_mask | CC_peaks |
|---|---|---|---|---|---|
| deposited | 4.83 | 0.494 | 0.716 | 0.788 | 0.614 |
| DeepEMhancer | 4.18 | 0.425 | 0.676 | 0.682 | 0.627 |
| phenix.auto_sharpen | 4.82 | 0.492 | 0.675 | 0.764 | 0.575 |
| **EMReady** | **3.57** | **0.542** | **0.855** | **0.798** | **0.753** |

The numbers in bold fonts indicate the best performances for the corresponding metrics.

## Evaluations on primary maps

We first evaluate the performance of EMReady on the test set of primary cryo-EM maps through the deposited map-model Fourier shell correlation (FSC) calculated by phenix.mtriage[43]. Here, the primary map indicates the final reconstruction result deposited in the EMDB that is usually post-processed, as opposed to the case of half-maps. It should be noticed that by default we calculate the unmasked FSC in this work, unless otherwise specified. Table 1 lists the unmasked map-model FSC-0.5 (FSC05) for the test set of 110 deposited primary cryo-EM maps. For comparisons, the table also shows the corresponding results for the deposited maps and the maps processed by DeepEMhancer[38] and phenix.auto_sharpen[29]. Only 100 out of 110 maps were successfully processed by DeepEMhancer. It can be seen from Fig. 2a, b that the maps processed by EMReady obtained significantly better FSC-0.5 compared to the deposited primary maps. On average, EMReady achieved a map-model FSC-0.5 of 3.57 Å, which is significantly improved from 4.83 Å for the deposited maps, 4.18 Å for the DeepEMhancer-processed maps, and 4.82 Å for the phenix.-auto_sharpen-processed maps. The detailed map-model FSC-0.5 values for each test case are listed in Supplementary Data 1.

Another important metric of map quality is the Q-score, which measures the resolvability of individual atoms in a cryo-EM map[44]. The Q-score can also be an indicator of map quality as its averaged value over the entire model correlates well with the reported resolution of the map. Here, the Q-score is calculated using the PDB structure built into the map. We report the average Q-scores of protein atoms for each of the maps in the test set. The average Q-scores for the entire test set are listed in Table 1. It can be seen from Fig. 2c that the maps processed by EMReady obtained a significantly higher average Q-score than the

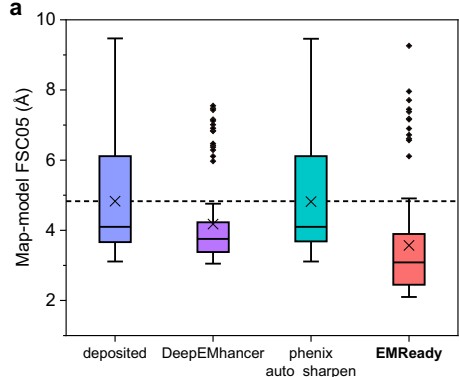

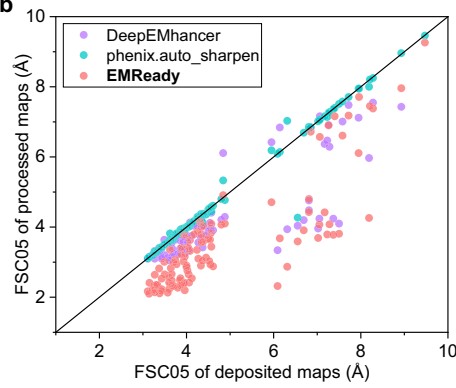

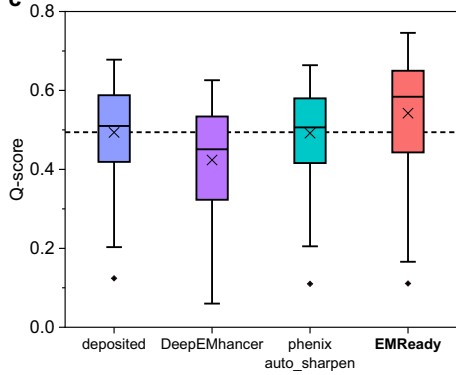

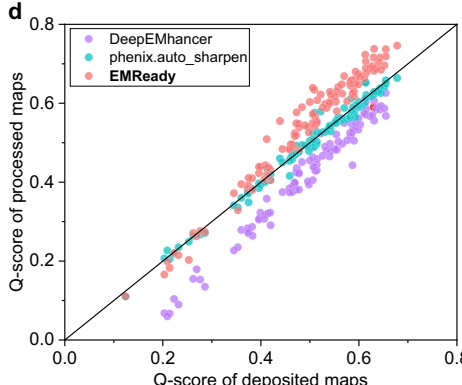

**Fig. 2 | Comparison of the unmasked map-model FSC-0.5 and Q-score on the test set of 110 deposited primary maps. a, c** Box-whisker plots of unmasked FSC-0.5 (**a**) and Q-score (**c**) for the deposited, DeepEMhancer-processed, phenix.-auto_sharpen-processed, and EMReady-processed maps ($n = 110$ individual test cases). The center line is the median, the cross is the mean, lower and upper hinges represent the first and third quartile, the whiskers stretch to 1.5 times the interquartile range from the corresponding hinge, and the outliers are plotted as diamonds. Dashed lines stand for the average values of deposited primary maps. **b, d** Comparison of unmasked FSC-0.5 (**b**) and Q-score (**d**) between the deposited and processed maps on each test case. Source data are provided as a Source Data file.

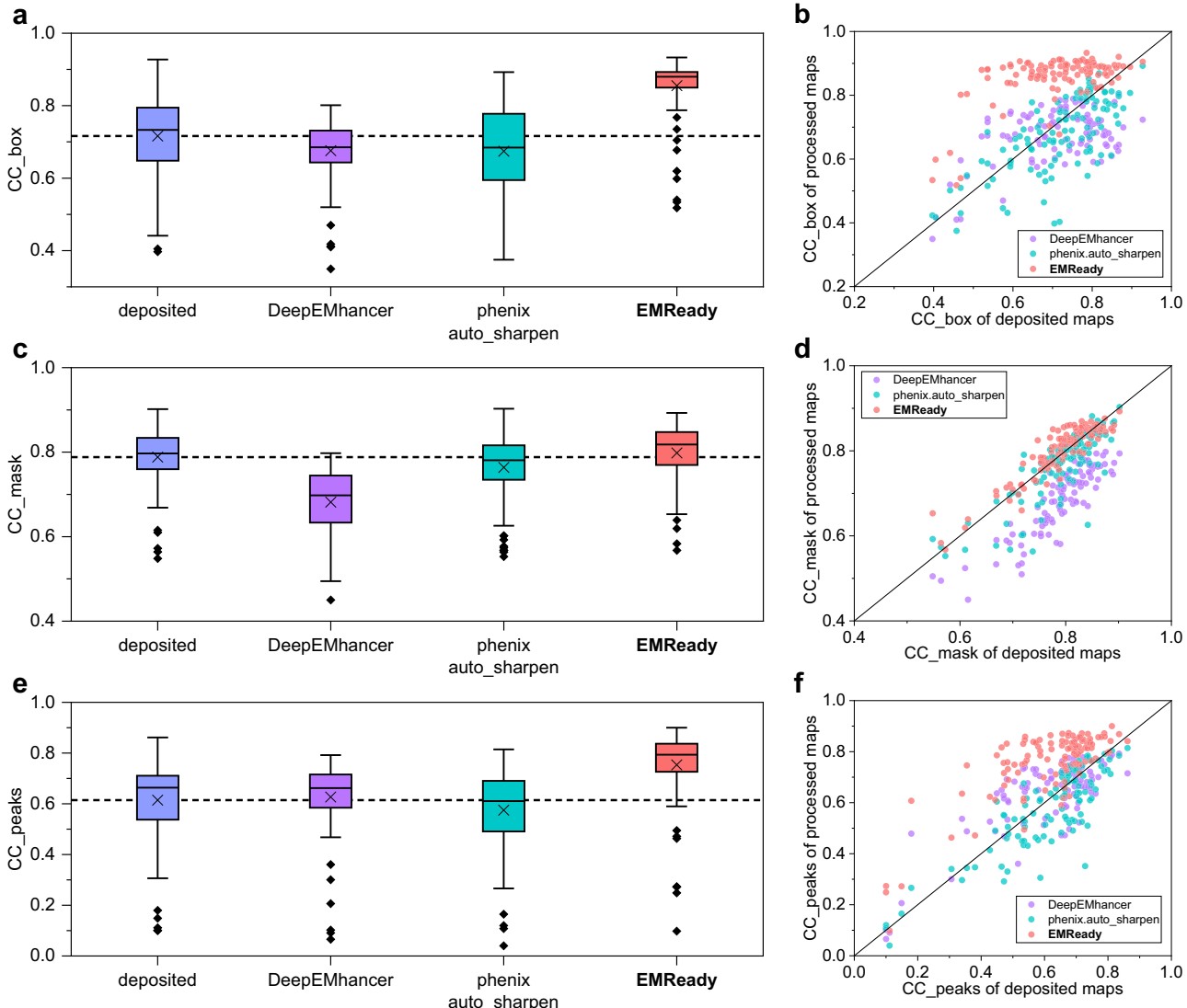

**Fig. 3 | Comparison of the CC values on the test set of 110 deposited primary maps. a, c, e** Box-whisker plots of CC_box (**a**), CC_mask (**c**), and CC_peaks (**e**) for the deposited, DeepEMhancer-processed, phenix.auto_sharpen-processed, and EMReady-processed maps ($n = 110$ individual test cases). The center line is the median, the cross is the mean, lower and upper hinges represent the first and third quartile, the whiskers stretch to 1.5 times the interquartile range from the corresponding hinge, and the outliers are plotted as diamonds. Dashed lines stand for the average CC values of deposited primary maps. **b, d, f** Comparison of CC_box (**b**), CC_mask (**d**), and CC_peaks (**f**) between the deposited and processed maps on each test case. Source data are provided as a Source Data file.

deposited maps and the maps processed by other methods. Specifically, EMReady achieved an average Q-score of 0.542, which is significantly higher than 0.494 for the deposited maps, 0.425 for DeepEMhancer, and 0.492 for phenix.auto_sharpen. As shown in Fig. 2d, EMReady improved the Q-scores for 96 out of 110 deposited maps on the test set. Specific Q-scores for each of the test cases are listed in Supplementary Data 1.

It should be noted that the Q-score as well as the map-model FSC are indirectly optimized during the training of EMReady on the simulated maps that are derived from PDB structures through a Gaussian forward model. Therefore, the improvement in Q-score and map-model FSC by EMReady here may be expected and would not be the best indicator for the improvement in map quality. Nevertheless, given that our test cases are independent from the training set under the threshold of <30% sequence identity, the Q-score and map-model FSC here would still be valuable metrics to measure the robustness and general applicability of EMReady across maps.

We further measure the performance of EMReady in terms of the correlation coefficient (CC) values between model-map density and experimental map density[43]. Three different CC metrics are reported, including CC_box, CC_mask, and CC_peaks. The measured CC values are listed in Table 1. As shown from Fig. 3a, c, e, EMReady yielded significant improvements on the deposited maps for all the CC values. The average CC_box, CC_mask, and CC_peaks values for the processed maps by EMReady are 0.855, 0.798, and 0.753, respectively, which are significantly higher than 0.716, 0.788, and 0.614 for the deposited maps. DeepEMhancer and phenix.auto_sharpen cannot improve the CC values on these deposited primary maps. As observed in Fig. 3b, d, f, compared to the deposited primary maps, the maps processed by EMReady give better CC_box, CC_mask and CC_peaks values. Specifically, EMReady has increased the CC_box values for 105 out of 110 maps, increased the CC_mask values for 75 out of 110 maps, and increased the CC_peaks values for 101 out of 110 maps. The detailed CC values for each of the 110 tested primary maps are listed in Supplementary Data 1.

Figure 4a shows a comparison between the deposited and EMReady-processed maps for EMD-22216, which is the mitochondrial calcium uniporter (MCU) holocomplex in low-calcium blocking state

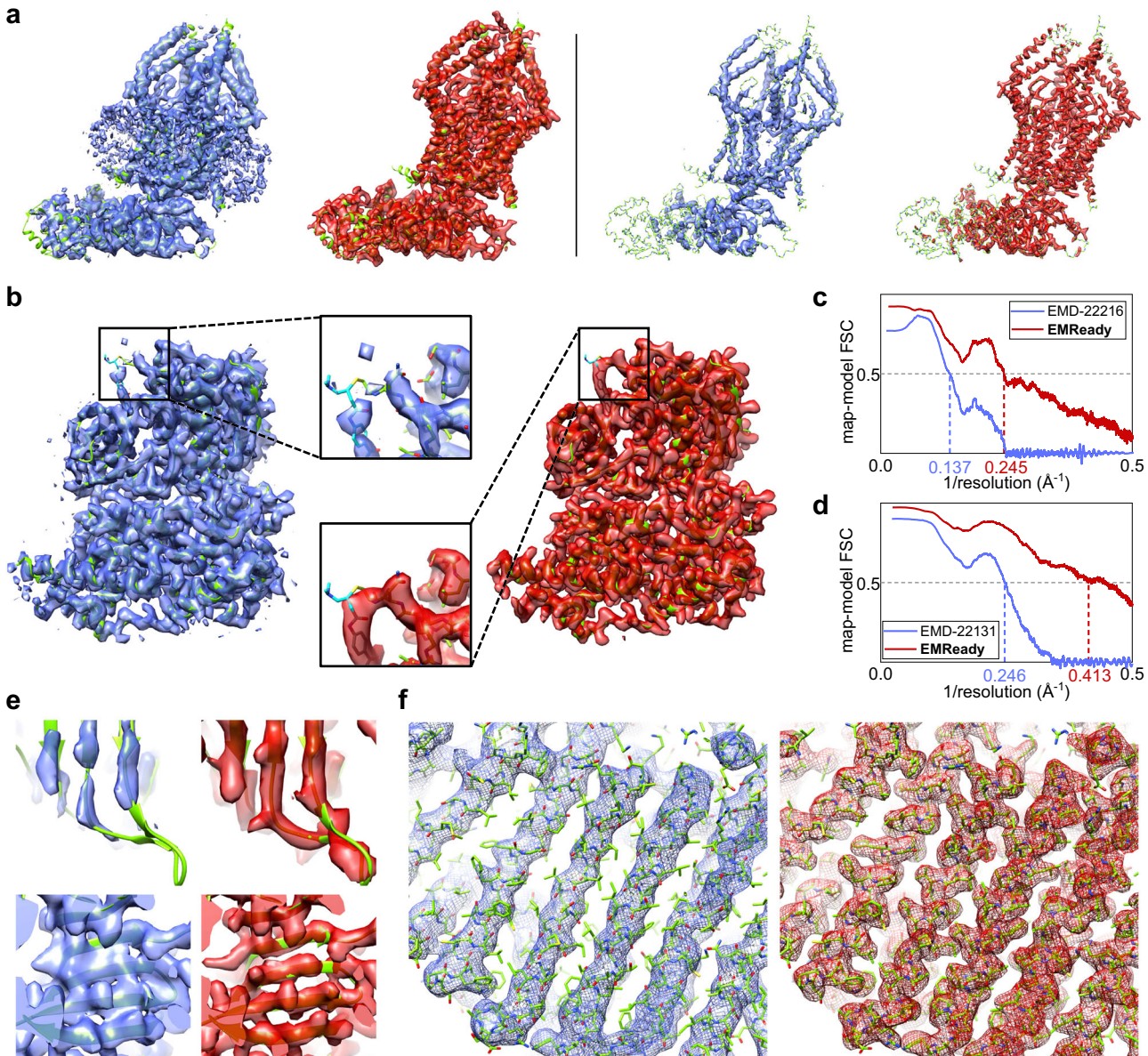

**Fig. 4 | Examples of the improved EM maps by EMReady.** The deposited primary maps are colored in blue, the EMReady-processed maps are in red, and the PDB structures are in green. **a** EMD-22216 (associated PDB ID: 6XJX) at 4.6 Å resolution, where the Left panel is for lower contour level and the Right panel is for higher contour level. **b** EMD-22131 (associated PDB ID: 6XD3) at 3.3 Å resolution. The enlarged views at the center compare the density regions around a ligand (Chemical ID: V0G). **c** Map-model Fourier shell correlation versus the inverse resolution for EMD-22216. **d** Map-model Fourier shell correlation versus the inverse resolution for EMD-22131. **e** EMD-10213 (associated PDB ID: 6SJ7) at 3.5 Å resolution, of which two different β-sheet regions are shown in the top and bottom rows, respectively. **f** EMD-0257 (associated PDB ID: 6HRA) at 3.7 Å resolution, where the left panel is for the average map of two half-maps and the right panel is for the EMReady-processed map. Source data are provided as a Source Data file.

(PDB ID: 6XJX). The left panel of Fig. 4a shows the comparison at a lower contour level. It can be seen from the figure that the transmembrane domain of MCU is buried in the lipid nanodisc in the deposited map. However, EMReady effectively improves the contrast of the region between the macromolecule and the lipid nanodisc, making the lipid region almost invisible in the processed map. In addition, EMReady also effectively reduces the background noise around the macromolecule. In order to reveal the improvement in the high-density regions of EM maps, we further compare the deposited and processed maps at a higher contour level. As shown in the right panel of Fig. 4a, the high-density regions in the processed map by EMReady perfectly fit the helical backbone traces of the PDB model, while the deposited map cannot provide such details. Correspondingly, the processed map by EMReady achieved an improved

unmasked FSC-0.5 and Q-score of 4.08 Å and 0.411, respectively, compared with 7.28 Å and 0.375 by the deposited primary map. The EMReady-processed map also yields improved CC_box and CC_peaks values of 0.863 and 0.731, respectively, which are significantly higher than 0.668 and 0.484 for the deposited map.

Figure 4b shows the example of EMD-22131, a 3.3 Å cryo-EM map of the human CDK-activating kinase (PDB ID: 6XD3). For this case, although EMD-22131 already has a good quality with an unmasked FSC-0.5 of 4.06 Å and a Q-score of 0.603, EMReady can further improve its quality and achieved a significantly better FSC-0.5 and Q-score of 2.42 Å and 0.668, respectively. Correspondingly, the CC_box, CC_mask, and CC_peaks of EMD-22131 are increased from 0.717, 0.808, and 0.674 for the deposited map to 0.912, 0.861, and 0.862 for the EMReady-processed map, respectively. In addition, the background

**Table 2 | Map quality on the test set of 25 pairs of half-maps**

| Method | FSC-0.5 (Å) | Q-score | CC_box | CC_mask | CC_peaks |
|---|---|---|---|---|---|
| half-maps | 5.17 | 0.399 | 0.763 | 0.729 | 0.626 |
| DeepEMhancer | 4.71 | 0.394 | 0.722 | 0.716 | 0.680 |
| LocScale | 5.04 | 0.409 | 0.685 | 0.686 | 0.688 |
| LocSpiral | 4.89 | 0.418 | 0.755 | 0.743 | 0.636 |
| phenix.auto_sharpen | 5.18 | 0.476 | 0.727 | 0.773 | 0.625 |
| phenix.resolve_cryo_em | 4.61 | 0.456 | 0.474 | 0.672 | 0.458 |
| **EMReady** | **4.07** | **0.491** | **0.873** | **0.794** | **0.760** |

The results of half-maps are calculated on the averaged map of two half-maps. The numbers in bold fonts indicate the best performances for the corresponding metrics.

noises in the deposited map are suppressed by EMReady, and the density signals for the macromolecule are enhanced. As observed in the enlarged view of Fig. 4b, the density signal for a ligand (Chemical ID: V0G) in the map is also improved by EMReady. The curves of unmasked map-model FSC versus the inverse resolution also demonstrate the improvement of the EMReady-processed map over the deposited map (Fig. 4c, d).

Figure 4e focuses two β-sheet regions of EMD-10213 that is associated with the structure of the human DDB1-DDA1-DCAF15 E3 ubiquitin ligase bound to RBM39 and Indisulam (PDB ID: 6SJ7). It can be seen from the figure that EMReady can improve the map density for the β-sheet region, not only by connecting the disconnected density fragments, but also by splitting an integrated region for a β-sheet into strips for individual β-strands. At the level of the entire map, EMReady improves the unmasked FSC-0.5 from 3.62 Å to 3.36 Å and improves the Q-score from 0.555 to 0.580. In addition, the EMReady-processed map achieved the CC_box, CC_mask, and CC_peaks values of 0.886, 0.824, and 0.792, respectively, which are significantly higher than 0.832, 0.798, and 0.693 for the deposited map.

In addition, we also give several examples of side-chains to show the improvement relative to the deposited map obtained by EMReady compared to other post-processing approaches including DeepEMhancer and phenix.auto_sharpen (Supplementary Fig. 1). As can be seen from the figure, EMReady provides improvements in both main-chain and side-chain densities for helices and sheets from maps of varied resolutions, compared to DeepEMhancer and phenix.auto_sharpen.

Besides the default EMReady model with a grid size of 1.0 Å, we also developed another EMReady model at 0.5 Å grid size to accommodate those cryo-EM maps with a small voxel size of <1.0 Å. Namely, during the training and evaluation of the 0.5 Å EMReady model, the grid size of the maps is unified to 0.5 Å. The EMReady model with 0.5 Å grid size is evaluated on a subset of 17 experimental maps that have a voxel size of below 1.0 Å. It is shown that overall the maps processed by EMReady at a grid size of 0.5 Å can also achieve significant improvements in terms of unmasked FSC-0.5, Q-scores, and CC values from the deposited maps (Supplementary Data 2). These results demonstrate the robustness and general applicability of EMReady.

## Evaluations on half-maps

In addition to primary cryo-EM maps, half-maps, which have nominally independent errors, are required as the input for most of the post-processing methods. Therefore, we further evaluated the performance of EMReady on a test set of 25 pairs of half-maps. The results are listed in Table 2 and shown in Figs. 5 and 6. We used the average map of two half-maps as the input of EMReady and also for the evaluation of half-maps. For comparison, the table and figures also give the unmasked map-model FSC-0.5 values of the average map of two half-maps and the maps processed by five other post-processing approaches including DeepEMhancer, LocScale, LocSpiral, phenix.auto_sharpen, and phenix.resolve_cryo_em (density modification). The evaluation

results for each of the test cases can be found in Supplementary Data 3. It can be seen from Table 2 that EMReady significantly improved the unmasked map-model FSC-0.5 of the half-maps and performed the best among the six post-processing methods (Fig. 5a). EMReady improved the unmasked FSC-0.5 for all of the 25 maps (Fig. 5b). On average, EMReady achieved an FSC-0.5 of 4.07 Å, compared with 5.17 Å for averaged half-maps.

Besides FSC-0.5, the Q-scores of EMReady-processed half-maps are also considerably improved (Fig. 5c). EMReady achieved an improved average Q-score of 0.491, which is higher than 0.399 for the averaged half-maps, 0.394 for DeepEMhancer, 0.409 for LocScale, 0.418 for LocSpiral, 0.476 for phenix.auto_sharpen, and 0.456 for phenix.resolve_cryo_em (density modification). Specifically, EMReady increased the Q-scores for 23 out of the total of 25 pairs of half-maps, as shown in Fig. 5d.

Furthermore, EMReady can also improve the CC_box, CC_mask, and CC_peaks values of averaged half-maps (Table 2). The processed maps by EMReady achieved the average CC_box, CC_mask, and CC_peaks values of 0.873, 0.794, and 0.760, respectively, which are significantly improved from only 0.763, 0.729, and 0.626 by the averaged half-maps and were also higher than those achieved by other post-processing approaches (Fig. 6a, c, e). As shown in Fig. 6b, d, f, the majority of the processed half-maps by EMReady obtained improved CC values. Specifically, out of the 25 pairs of half-maps, EMReady improved the CC_box for 22 maps, improved the CC_mask for 20 maps, and improved the CC_peaks for 24 maps.

Figure 4f shows an example of how EMReady improves the quality of half-maps. It can be seen from the figure that although the backbone traces of α-helices can be clearly seen in the original averaged half-maps of EMD-0257, few density signals of side chains can be observed. However, after being processed by EMReady, the density map shows legible density signals for side-chains. In addition, the improved side-chain signals have correct orientations and suitable sizes. Quantitatively, EMReady improved the unmasked map-model FSC-0.5 and Q-score for the averaged half-maps from 4.32 Å and 0.369 to 3.28 Å and 0.519, respectively. In addition, the CC_box, CC_mask, and CC_peaks were also considerably improved from 0.709, 0.736, and 0.573 to 0.919, 0.849, and 0.844.

In the above evaluations, we have demonstrated that the EMReady-processed maps obtained an increased similarity to a gaussian forward model derived from the PDB structure. However, this might not be the true underlying signal that is observed in cryo-EM[45]. Therefore, to evaluate how EMReady truly improves the quality of the reconstruction itself, we conducted an additional experiment. Specifically, for each case in the test set of 25 pairs of half-maps, we applied EMReady to one of the two half-maps and calculated the unmasked FSC between one processed half-map and the other unprocessed half-map. The evaluation results are shown in Supplementary Fig. 2a, b. It can be seen from the figure that the FSC-0.5 values between one processed half-map and the other unprocessed half-map are significantly improved. On average, the FSC-0.5 between half-maps is improved from 5.60 Å for the unprocessed half-maps to 5.22 Å using EMReady-processed half-map #1 and 5.18 Å using EMReady-processed half-map #2. These results suggest that EMReady indeed captures the true underlying signal from the half-map and injects it to the output processed map. As illustrated in Supplementary Fig. 2c, EMReady captures clear traces of the β-sheets from both half-maps of EMD-0071. The FSC curves between one processed half-map and the other unprocessed half-map are continuously enhanced by EMReady over a long range of inverse resolutions, compared with the case between two unprocessed half-maps (Supplementary Fig. 2d). Accordingly, EMReady achieves improved FSC-0.5 values of 4.95 Å and 4.88 Å for the half-maps with the processed half-map #1 and half-map #2, respectively, compared with 5.55 Å between the unprocessed half-maps. The detailed experiment results are listed in Supplementary Data 4.

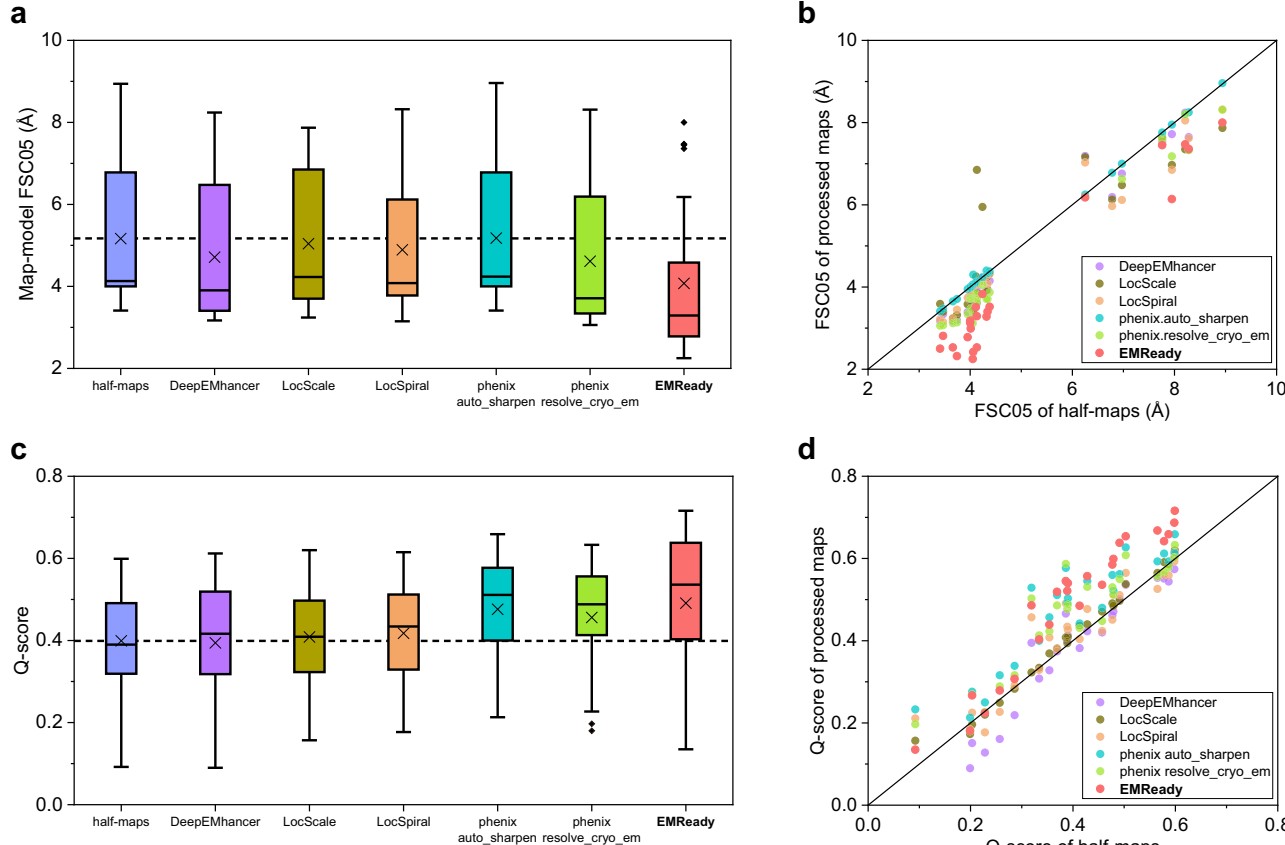

**Fig. 5 | Comparison of the unmasked map-model FSC-0.5 and Q-score on the test set of 25 pairs of half-maps. a**, **c** Box-whisker plots of unmasked map-model FSC-0.5 (**a**) and Q-score (**c**) for the average maps of half-maps and the maps processed by DeepEMhancer, LocScale, LocSpiral, phenix.auto_sharpen, phenix.resolve_cryo_em (density modification), and EMReady (n = 25 individual test cases). The center line is the median, the cross is the mean, lower and upper hinges represent the first and third quartile, the whiskers stretch to 1.5 times the inter-quartile range from the corresponding hinge, and the outliers are plotted as diamonds. Dashed lines stand for the average values of deposited half-maps. **b**, **d** Comparison of unmasked FSC-0.5 (**b**) and Q-score (**d**) between the deposited and processed maps on each test case. Source data are provided as a Source Data file.

## Improvement in map interpretability

As the ultimate goal of cryo-EM is to determine the atomic model from the EM map, the true improvement in processed maps should be reflected in the improvement in the built models, that is, the improvement in map interpretability. Therefore, we further evaluate the performance of EMReady in terms of de novo model building on a test set of 682 chains of cryo-EM primary maps. Specifically, we use phenix.map_to_model[17–19] to automatically build the atomic models from the map regions segmented within 4.0 Å from each of the chains. The built models are compared with the deposited PDB structures through phenix.chain_comparison. Two metrics, residue coverage and sequence match percentages, are reported as the measure of model quality. The evaluation results on the test set are shown in Fig. 7a, b and listed in Supplementary Table 1. For comparison, the figures also display the corresponding results of deposited maps, DeepEMhancer-processed, and phenix.auto_sharpen-processed maps. The evaluation results for each test case can be found in Supplementary Data 5. It can be seen from the figure that EMReady significantly improved the residue coverage of built atomic models and gave an average value of 79.7%, which is significantly higher than 64.2% for the deposited maps, 58.1% for DeepEMhancer, and 64.3% for phenix.auto_sharpen. Similar trends can also be found in the sequence match percentages of built models. The atomic models built from the EMReady-processed maps achieved a much higher sequence match of 50.4%, compared with 31.9% for the deposited models, 34.6% for DeepEMhancer, and 32.9% for phenix.auto_sharpen.

Supplementary Fig. 3a compares the atomic models built from the deposited map and from the EMReady-processed map (PDB ID: 5LZP, chain L). As indicated by the arrows, the deposited map (EMD-4128) suffers from heterogeneity in density signals, thus results in skipped backbone traces and promiscuous side-chain assignment. In contrast, the corresponding weak signals are significantly enhanced in the EMReady-processed map. Therefore, the EMReady-processed map leads to a better model built by phenix.map_to_model in both accurate backbone tracing and side-chain assignment. Specifically, the atomic model built from the EMReady-processed map achieved a residue coverage of 97.7% and a sequence match of 98.6%, compared with a residue coverage of 51.8% and a sequence match of 41.7% for the deposited map. Supplementary Fig. 3b gives another example for the chain h of EMD-10045 (PDB ID: 6RWX). Compared to the deposited map, the EMReady-processed map also significantly improved the continuity of density volume (indicated by an arrow), the side-chain packing and thereby the interpretability. The model built by phenix.map_to_model from the EMReady-processed map achieved a residue coverage of 95.5% and a sequence match of 89.3%, respectively, compared with only 72.3% and 60.2% for the deposited map.

To evaluate the general applicability of EMReady in improving map interpretability, we have also used another modeling algorithm, MAINMAST[20], to build the atomic models from the deposited, DeepEMhancer, phenix.auto_sharpen, and EMReady-processed maps on the test set of 385 protein chains. As shown in Fig. 7c, d, the MAINMAST models built on the EMReady-processed maps yielded a significantly

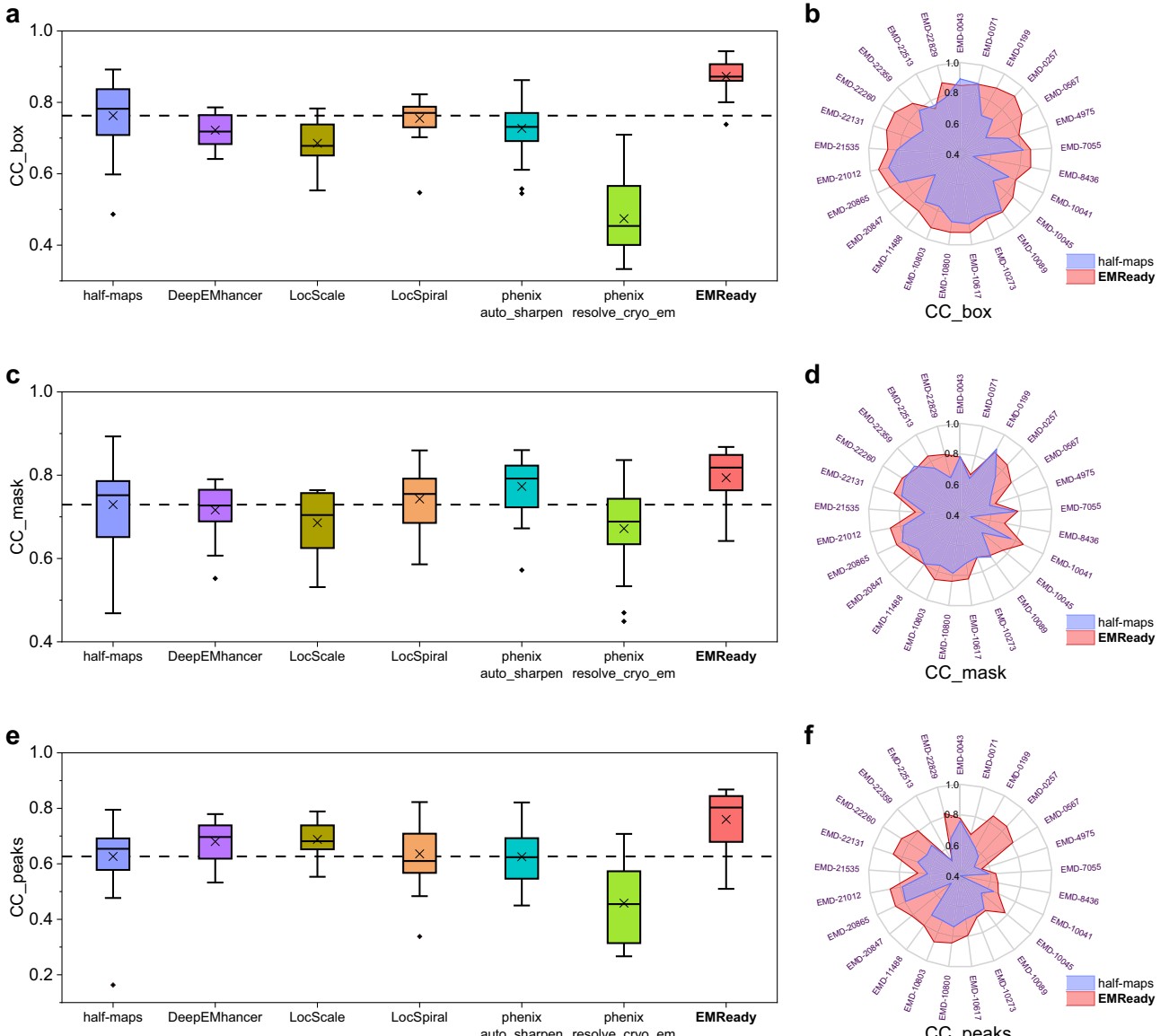

**Fig. 6 | Comparison of the CC values on the test set of 25 pairs of half-maps.**
**a**, **c**, **e** Box-whisker plots of CC_box (**a**), CC_mask (**c**), and CC_peaks (**e**) for the average maps of half-maps, the maps processed by DeepEMhancer, LocScale, LocSpiral, phenix.auto_sharpen, phenix.resolve_cryo_em (density modification) and EMReady (n = 25 individual test cases). The center line is the median, the cross is the mean, lower and upper hinges represent the first and third quartile, the whiskers stretch to 1.5 times the interquartile range from the corresponding hinge, and the outliers are plotted as diamonds. Dashed lines stand for the average CC values of deposited half-maps. **b**, **d**, **f** Comparison of CC_box (**b**), CC_mask (**d**), and CC_peaks (**f**) between the deposited map and EMReady-processed map on each test case. Source data are provided as a Source Data file.

improved coverage of 85.6%, compared with 73.9% for the deposited maps, 72.1% for DeepEMhancer, and 74.0% for phenix.auto_sharpen (Supplementary Table 1). The improvement of map interpretability can also be observed in sequence match. Specifically, EMReady obtained an average sequence match value of 33.8%, which is significantly higher than 15.2% for deposited maps, 14.6% for DeepEMhancer, and 15.4% for phenix.auto_sharpen (Supplementary Table 1). The evaluation results for each of the chains can be found in Supplementary Data 6. These results again demonstrate the improvement of map interpretability by EMReady.

**Evaluation against higher-resolution structures and maps**
Using the map-associated PDB structures as the reference structures, we have demonstrated the better quality of EMReady-processed maps than deposited maps. However, if the quality of a cryo-EM map is low, the associated PDB structure may contain errors, which would introduce biases into the evaluation results. Therefore, to further evaluate the robustness of EMReady, we used EMReady to process lower-resolution cryo-EM maps and then evaluated the EMReady-processed maps against higher-resolution PDB structures and maps.

The example of a cryo-EM map at 3.1 Å resolution for human apoferritin (EMD-20028) is shown in Fig. 8a, b. The reference PDB structure is built into a 1.8 Å EM map (EMD-20026) based on the X-ray structure of human apoferritin at 1.52 Å resolution (PDB ID: 3AJO)[44,46]. Since the voxel size of EMD-20028 is 0.65 Å, the EMReady model with 0.5 Å grid size is applied instead of the primary 1.0 Å model. As shown in Fig. 8a, the values of Fourier shell correlation between the map and the reference model are consistently improved by EMReady over a long range of inverse resolutions. It can be seen from Fig. 8b that the EMReady-processed map shows more high-resolution details in both backbone regions and side-chain regions than the original 3.1 Å map. Quantitatively, the unmasked map-model FSC-0.5 and Q-score

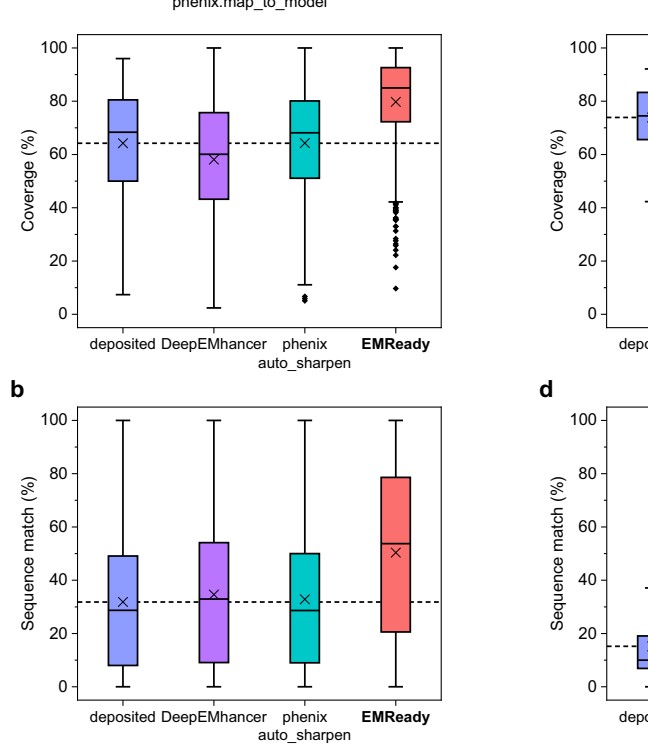

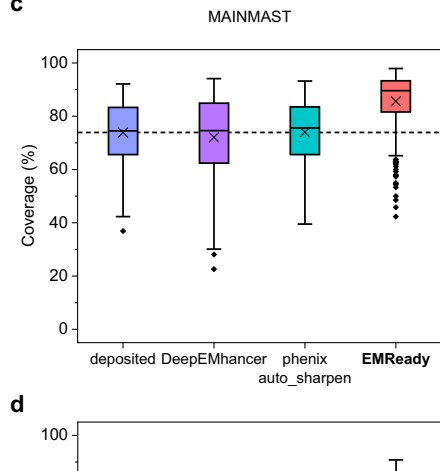

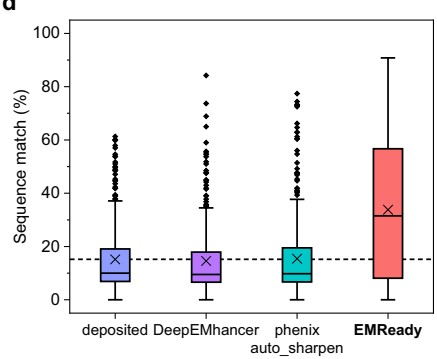

**Fig. 7 | Improvement in map interpretability. a, b** Box-whisker plots of residue coverage percentage (**a**) and sequence match percentage (**b**) for the models built by phenix.map_to_model for the deposited, DeepEMhancer-processed, phenix.-auto_sharpen-processed and EMReady-processed maps on the test set of *n* = 682 individual chains. **c, d** Box-whisker plots of residue coverage percentage (**c**) and sequence match percentage (**d**) for the models built by MAINMAST for the deposited, DeepEMhancer-processed, phenix.auto_sharpen-processed and

EMReady-processed maps on the test set of *n* = 385 individual protein chains. For the box-whisker plots, the center line is the median, the cross is the mean, lower and upper hinges represent the first and third quartile, the whiskers stretch to 1.5 times the interquartile range from the corresponding hinge, and the outliers are plotted as diamonds. The dashed lines represent the mean values of deposited maps. Source data are provided as a Source Data file.

between the map and the reference structure was considerably improved from 6.26 Å and 0.624 to 2.00 Å and 0.747, respectively. In addition, the CC_box, CC_mask, and CC_peaks of EMReady-processed map were 0.881, 0.828, and 0.839, respectively, which are higher than 0.720, 0.824, and 0.686 by the deposited map, respectively.

Similar improvement trends can also be observed on another cryo-EM map of human γ-secretase, EMD-2677 at 4.5 Å resolution (Fig. 8c, d). Here, the reference PDB structure is built from a cryo-EM map at 3.4 Å resolution (EMD-3061, associated PDB ID: 5A63). It can be seen from the figures that EMReady significantly improved the quality of the lower-resolution cryo-EM map, and the values of Fourier correlation between the map and the reference model were also consistently improved by EMReady. Correspondingly, the unmasked FSC-0.5 and Q-score are improved by EMReady from 7.54 Å and 0.277 to 4.72 Å and 0.297, respectively. The EMReady-processed map also achieved higher CC_box, CC_mask, and CC_peaks of 0.824, 0.668, and 0.686, respectively, compared to 0.504, 0.628, and 0.311 for the deposited map.

In addition to the evaluations against higher-resolution PDB structures, we also assessed our EMReady-processed maps using higher-resolution cryo-EM maps as the references. Specifically, we investigated the unmasked map-map FSCs of a lower-resolution map before and after applying EMReady against its corresponding higher-resolution map. The map-map FSCs are calculated by phenix.mtriage[43]. As shown in Supplementary Fig. 4a, b, the map-map FSC curves of EMD-20028 against EMD-20026 and EMD-2677 against EMD-3061 before and after applying EMReady are similar to their map-model counterparts (Fig. 8a, c). These results further confirm the improvement of the map quality in the EMReady-processed maps.

Moreover, we also tested EMReady on a much lower resolution map of human γ-secretase, EMD-2678 at 5.4 Å resolution, which was reconstructed from a small number of 37310 particles. As a comparison, its corresponding 3.4 Å map EMD-3061 was reconstructed from 159549 particles. It is shown that the Fourier correlation of EMD-2678 against EMD-3061 is also improved over a very long range of inverse resolutions (Supplementary Fig. 4c). Comparing the above three cases also reveals that the map improvement by EMReady may to some extent depend on the resolution (Supplementary Fig. 4a–c). Such trend can be partially attributed to the fact that EMReady tends to be more confident to modify the densities in a map with higher resolution and reliable density information. In contrast, if a map has a low resolution like the case of EMD-2678, EMReady will try to maintain the original density signals in the map (Supplementary Fig. 4d). These results suggest that EMReady is reliable and robust in processing the maps with different resolutions, instead of just learning the simple mapping from experimental maps to simulated maps.

### Validation of density modifications by EMReady

Unlike traditional map sharpening methods that aim at optimizing the resolution-dependent weighting of the amplitude components of Fourier coefficients in the Fourier representation of cryo-EM maps, EMReady directly modifies the input map in real space and would change the phases and amplitudes. Therefore, it is of pivotal importance to evaluate whether the density modifications by EMReady are reliable. However, a direct validation for the density modification is impossible because the absolute ground-truth maps/models are not available in real-world scenarios. Instead, we have adopted an indirect validation way by extensively examining the

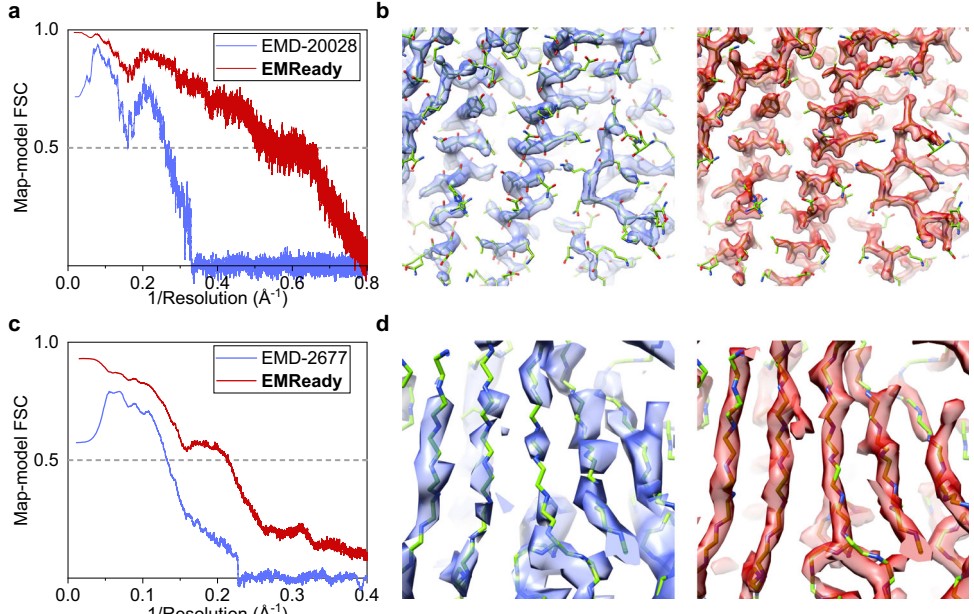

**Fig. 8 | Applying EMReady to lower-resolution maps and evaluating against higher-resolution PDB structures. a, b** Evaluation of an apoferritin 3.1 Å cryo-EM map (EMD-20028) using an apoferritin 1.52 Å X-ray structure (PDB ID: 3AJO) built into a 1.8 Å cryo-EM map (EMD-20026). **c, d** Evaluation of a 4.5 Å cryo-EM map (EMD-2677) for human γ-secretase using a 3.4 Å reference structure (PDB ID: 5A63; EMD-3061). **a, c** The unmasked map-model Fourier shell correlation curves versus the inverse resolution. **b, d** Comparison of the density volumes between the deposited maps (blue) and EMReady-processed maps (red). The higher-resolution reference PDB structures are colored in green. Source data are provided as a Source Data file.

reasonability of the density modification in the processed maps by EMReady in various cases.

The first case is the primary map of EMD-0257 at 3.7 Å resolution in which the N and A cytoplasmic domains of KdpB are more flexible and thus have a lower resolution than the transmembrane helices in the deposited map (Supplementary Fig. 5a). Here, the local resolution of the deposited map is calculated by MonoRes[37]. To evaluate whether the density modification by EMReady is reliable, we calculated the local correlations of the deposited map (Supplementary Fig. 5b) and the EMReady-processed map (Supplementary Fig. 5c) against the PDB map simulated from the associated structure (PDB ID: 6HRA) (Supplementary Fig. 5d). Here, a higher correlation means a better similarity with the simulated PDB map, and a lower correlation stands for a higher difference from the PDB map. Several notable properties can be observed from the figures. First, the regions with the lowest correlation (i.e., the highest difference) between the EMReady-processed map and the simulated PDB map tend to be located in the low resolution/flexible KdpB parts, whereas the regions with a higher correlation (i.e., a better similarity) between the EMReady-processed map and the simulated PDB map tend to be located in the higher resolution/more reliable transmembrane part (Supplementary Fig. 5c). These trends are consistent with those in the deposited map (Supplementary Fig. 5b). Second, two notable features can be seen by comparing the EMReady-processed map (Supplementary Fig. 5c) and the deposited maps (Supplementary Fig. 5a, b). One feature is that EMReady improves the local correlation (i.e., give a better similarity) with the simulated PDB map in the more reliable transmembrane region with local resolutions of about 3–6 Å, but remains a low-density correlation with the PDB map in the flexible KdpB parts with local resolutions of >6 Å. In other words, EMReady tends to modify the density towards the PDB structure for the higher resolution/more reliable regions, but does not try to overfit the density towards the lower resolution/more flexible parts, which are wanted in terms of map reliability. This phenomenon may be understood because EMReady is trained on the cases with 3–6 Å resolutions and thus tends to let the lower-resolution regions remain intact. The other feature is that EMReady succeeds in suppressing the

noises of lipid solvents around the protein without removing the protein density signals or adding artifacts from noises (Supplementary Fig. 5c). These results demonstrate the reasonability of the density modification by EMReady. Correspondingly, the EMReady-processed map achieved improved FSC-0.5, Q-score, CC_box, CC_mask, and CC_peaks of respectively 2.90 Å, 0.573, 0.900, 0.844, and 0.837, compared with 4.33 Å, 0.506, 0.697, 0.827, and 0.572 of the deposited map.

The second case is EMD-5447, for which the particles were selected from very noisy micrographs using an external reference and thus might have resulted in an overfitted reconstruction[47,48]. Supplementary Fig. 5e compares the deposited map and the EMReady-processed map of EMD-5447. It can be seen from the figure that EMReady almost does no modification to the map. Such finding may be partially due to that the density patterns in this overfitted reconstruction are not similar to those seen in the training set of EMReady. This is encouraging because EMReady does not overfit an improperly reconstructed map towards structure-like patterns.

We further evaluated the impact of noises in input EM density map on the output of EMReady. Specifically, the simulated map of a PDB structure (PDB ID: 6OOH) was created at the reported resolution using a grid step of 1.0 Å. After normalizing the simulated map to the range of 0.0–1.0, the Gaussian noise with a standard deviation of 1/6 was added using Xmipp[49]. Then, EMReady was used to process the noisy simulated map. As shown in Supplementary Fig. 6a, EMReady did not misinterpret the noises as the signals and successfully recovered the density volume of the simulated map by suppressing the noises in the noisy map. The unmasked map-map FSC curves of the noisy map and the EMReady-processed map against the simulated map are displayed in Supplementary Fig. 6b. It can be seen from the figure that compared with the noisy map, the unmasked map-map FSC of the EMReady-processed map is consistently improved over a long range of inverse resolutions. Similar trend is also witnessed in the unmasked map-model FSC curves (Supplementary Fig. 6c). It is worth mentioning that this is just an extreme example to illustrate how EMReady deals with very noisy cryo-EM maps, while we have demonstrated that

EMReady is also capable of handling the real noises observed in experimental cryo-EM maps (Fig. 4a, b and Supplementary Fig. 4d).

We then investigated the behavior of EMReady on realistic noises. Displayed in Supplementary Fig. 6d is the same example as that in Fig. 4a but at much lower contour thresholds. It can be seen from the figure that the noises introduced by the lipid nanodiscs are well suppressed and only appear in the EMReady-processed map at very low thresholds. More importantly, one can see that the noises processed by EMReady do not seem to add significant structural artifacts to the density of macromolecule, compared with those in the deposited map.

Finally, we examined the density modification of EMReady on a map with pure noise. Following the same procedure as the previous experiment, Xmipp was used to add the Gaussian white noise with a standard deviation of 1/6 to an empty grid. As shown in Supplementary Fig. 6e, EMReady successfully suppressed the Gaussian white noise, which can only present at very low contour levels compared to the typical contour level required to enclose the macromolecule in EMReady-processed experimental maps (e.g., threshold of 2.0 in Supplementary Fig. 6d). Moreover, despite some agglomeration of noise, the resulting EMReady-processed map maintained the overall form of Gaussian noise. Although we notice that the agglomeration of noise may lead to some structural patterns (Supplementary Fig. 6e), the density values for structured noises are extremely low in the EMReady-processed map compared to those for the underlying macromolecule (Supplementary Fig. 6a, d). For example, structured noises can be seen at density thresholds of 0.05 and 0.1, which approximately correspond to density values of respectively 1/6 and 1/3 in the original noise map (calculated by rescaling the EMReady-processed map to have an equal standard deviation). These findings suggest that EMReady successfully recognizes the texture of noises, and thus produces an output with minimal artifacts. Nevertheless, it should be emphasized that low density values are risky and should be carefully handled by potential users when interpreting the EMReady-processed map.

The findings on the above cases demonstrate the robustness and reliability of the density modification by EMReady. Namely, the modification of EMReady is based on the local quality and resolvability of a map. Specifically, in addition to efficiently suppressing the background noise in a map, EMReady modifies the density map based on both global resolution and local environment. On one hand, EMReady is effective to add high-resolution details for higher-resolution or rigid regions. On the other hand, EMReady would not forcibly overfit the density signals towards the PDB structure for the lower-resolution or intrinsically flexible regions that lack adequate and valid structural context.

## Ablation experiments

There are two major differences between EMReady and similar deep learning methods: One is the extraction of non-local features within the size of the input density slices ($48 Å \times 48 Å \times 48 Å$) by learning strategies including SCUNet with swin transformer as well as the SSIM loss; The other is the use of simulated map as the learning target. To investigate how the non-local components of our deep learning framework affect the performance of EMReady, we conducted extensive ablation experiments of our EMReady framework. Specifically, we trained a total of four additional ablation models of EMReady, including two EMReady models with different input box sizes, one EMReady model using the local UNet++[50] network, and one EMReady model without using the non-local SSIM loss.

We first calculated the unmasked FSC-0.5, Q-score, and CC values of these ablation models on the test set of primary maps and on the test set of half-maps. The detailed ablation results on each test case of the test set of 110 primary maps and the test set of 25 half-maps are listed in Supplementary Data 7 and 8, respectively. The average evaluation results are presented in Supplementary Table 2. As shown in the table, the performance of the model without using SSIM loss is significantly worse than the baseline model on both test sets in terms of different evaluation metrics. This suggests that the proposed non-local SSIM loss is important for improving the accuracy of our EMReady model. Moreover, the UNet++ model also performed significantly worse than the baseline model in terms of FSC-0.5 and Q-score on both test sets, as shown in Supplementary Fig. 7. These results demonstrate that the performance of EMReady indeed benefits from the non-local components in our deep learning framework, though the non-locality of EMReady is limited to the size of input density slices of $48 Å \times 48 Å \times 48 Å$. To investigate the impact of input box size on EMReady, we compared the results of the EMReady models with different input box sizes. It is found that using a larger input box is beneficial for the performance of EMReady. As the range of non-locality in EMReady is directly proportional to the size of input density box, the better performance with a larger input box can be attributed to its longer range of non-locality. These ablation results highlight the necessity of including SSIM loss, adopting SCUNet architecture, and using a large input box in EMReady.

We further compared the learning curves of the baseline model with the UNet++ model and the model trained without SSIM loss (Supplementary Fig. 8a–d). The detailed loss values are listed in Supplementary Data 9. It can be seen from the figure that the baseline model has lower smooth L1 loss and SSIM loss, compared with the local UNet++ model. Moreover, the smooth L1 loss of the model trained without SSIM is significantly lower than that of the baseline model and the UNet++ model on both the training and validation set. This means that the model trained without SSIM loss tends to yield an overfitted result with local distance as low as possible to its learning target while ignoring the important structural correlation in the density signals, thus resulting a worse performance compared with the baseline model on the independent testing set. Supplementary Fig. 8e displays an example of how the SSIM loss prevents the model from overfitting on the primary map of EMD-11231 at 4.3 Å resolution. The baseline model achieved better FSC-0.5 and Q-score of 4.07 Å and 0.483, compared with 4.31 Å and 0.467 by the UNet++ model, and 4.11 Å and 0.453 by the model without SSIM loss. In addition, one can also see that compared with the maps processed by the baseline model and the UNet++ model, the density volume for a part of the Nqo12 subunit was mistakenly filtered out after being processed by the model trained without SSIM loss. This suggests that incorporating the SSIM loss into the training process can effectively avoid such mis-modification that is caused by ignoring structural correlation in the overfitted model.

The direct ablation experiment of the simulated PDB maps for EMReady is difficult because EMReady cannot take the LocScale-processed maps that have been used by DeepEMhancer as the training set, because LocScale is only designed for unfiltered and unsharpened maps only. DeepEMhancer also cannot be trained on the training set of primary maps that are used by EMReady, because DeepEMhancer is designed to use half-maps for training. Nevertheless, the importance of the simulated maps may be roughly indicated by the difference between the results of EMReady and LocScale on the test set of 25 pairs of half-maps. Comparing Table 2 and Supplementary Table 2 reveals that the use of the simulated maps seems to have the most impact on the improvement of EMReady, suggesting the necessity of using simulated maps instead of LocScale-processed maps. However, as we discussed above, simply using the simulated maps with a smooth L1 loss is problematic because it will result in overfitting. Only using the simulated maps with a combination of smooth L1 loss and SSIM loss can effectively prevent the training from overfitting, and thus obtain a robust and reliable model, which is just another important point of our EMReady method.

## Discussion

In this study, we propose EMReady, a powerful deep learning framework to improve the interpretability of cryo-EM maps based on a three-

dimensional Swin-Conv-UNet architecture. EMReady addresses the critical challenges of loss of contrast and heterogeneity in cryo-EM maps using twofold local and non-local strategies. First, in the network architecture, EMReady adopts a swin convolutional block to incorporate the local modeling ability of residual convolutional layer and non-local modeling ability of swin transformer. The implemented multi-scale UNet can further enhance the local and non-local modeling ability. Second, in the loss function, EMReady not only measures the local difference through the smooth L1 distance, but also considers the non-local correlation effect through the SSIM between the processed experimental and simulated target maps. Compared with training with simple smooth L1 loss, incorporating SSIM loss in the training process can effectively prevent EMReady from possible overfitting.

Despite the powerfulness of non-local modeling in EMReady, it should be noted that the non-local range of EMReady is not global but have a limited distance because of the high memory cost of swin transformer during the training of EMReady. Namely, the size of input density box ($48\,\text{Å} \times 48\,\text{Å} \times 48\,\text{Å}$ in this study), which defines the non-local modeling range of EMReady, is strongly limited by the total memory of GPUs during the training, though the running of the trained EMReady model is not memory expensive and can be accommodated on a GPU with a memory as low as 8 GB. It is expected that the non-local modeling capability of EMReady can be further improved with more advanced GPUs with higher memory.

In addition to the local and non-local design in the network architecture, another important contribution to the performance of EMReady is the use of the simulated maps as the target maps during the training process. Although EMReady is benefited from the pure density signals in simulated maps, the simulated maps should be used with caution. First, the PDB structure may contain poorly modeled regions and thus introduce possible biases to the simulated maps. Addressing this issue, strict criteria of model quality and model-map correlation should be used when collecting the training cases, as we have done for EMReady. Second, our simulated map is calculated using a uniform resolution over the entire structure (Eq. 1), which ignores the heterogeneity in real experimental maps. Although we have shown that such discordance would not lead to an overfitting of EMReady, it may to some extent cause a sub-optimal performance of the trained model. It is expected that the deep learning model would be further improved if we take the structure and map heterogeneity into consideration, e.g., calculating the simulated map according to the B-factor of each atom, which will be left to our future study. Third, some special structure components like ligands, glycans, post-translational modification, and lipids may not be necessarily modeled in the PDB structures by their authors or have too few samples in the PDB structures. As such, the density information for such components may not be efficiently learned from the corresponding simulated maps.

Although EMReady has been trained on primary maps that are usually masked and/or sharpened, it is able to improve both raw half-maps and processed primary maps, whereas existing amplitude-reweighting-based sharpening methods are often designed to only process unmasked and unsharpened half-maps. Therefore, EMReady can be used to further improve the results of other post-processing methods through de-noising and density modification. However, unlike traditional sharpening methods which only modify the Fourier amplitude, EMReady directly modifies the map density in real space and would change both the phases and amplitudes. Therefore, although we have shown that the density modifications by EMReady are reliable on most of the test cases, users should still pay special attentions when analyzing the results of EMReady.

In addition, comparison of the EMReady results on unprocessed half-maps and post-processed primary maps reveals that EMReady performs better on the post-processed primary maps than on the unprocessed half-maps (Supplementary Fig. 9). Specifically, the average unmasked FSC-0.5 and Q-score for EMReady-processed primary maps are 3.81 Å and 0.536, compared to 4.07 Å and 0.491 for EMReady-processed half-maps, respectively. These results can be understood because EMReady is trained on post-processed primary maps. It is expected that EMReady can be further improved through training on unprocessed half-maps when more and more half-maps become available for their PDB structures. Before then, for the best performance of EMReady on unprocessed raw half-maps, users may try to apply a global B-factor-based sharpening method like RELION or phenix.auto_sharpen before applying EMReady.

Finally, it should be noted that despite the superior performance of EMReady, users should not solely rely on the EMReady-processed map when building a model from a cryo-EM map. As mentioned above, EMReady may not be able to properly handle some special structure components like small ligand, glycan, post-translational modification, lipid, etc. because they are under-represented in the training set. For those uncommon structures like ligands and glycans, EMReady may not be able to give significant density improvement (Supplementary Fig. 10a). For lipid molecules, EMReady tends to filter out their density signals (Supplementary Fig. 10b). In addition, for those density regions with extremely weak signals, EMReady may mis-recognize them as background noises and would suppress their density signals (Supplementary Fig. 10c), leading to incorrect modeling (Supplementary Data 5). Therefore, it is recommended that users try different post-processing methods and examine their modeling results by orthogonal ways. All in all, given the accuracy and robustness of EMReady in improving the quality and interpretability of cryo-EM maps, it is anticipated that EMReady will serve as a valuable tool for improving experimentally solved cryo-EM maps and thus help determine the atomic structures.

## Methods

### Network architecture

We use a Swin-Conv-UNet (SCUNet)[40] architecture to post-process cryo-EM density maps. Figure 1c shows the schematic of our deep learning architecture. EMReady consists of three encoder, one bottleneck, and three decoder swin-conv (SC) blocks with skip connections between encoders and decoders. Each SC block includes a swin transformer (SwinT) block for non-local modeling[41] paralleled to a residual convolution (RConv) block for local modeling. The window size of the swin transformer is set to 3. The 3D convolution layer with kernel size and stride of 2 is used as down-sampling, and the 3D transposed convolution layer with kernel size and stride of 2 is used as up-sampling. The details of the network architecture can be found in Supplementary Data 10. The inputs of our network are density slices of size $48 \times 48 \times 48$ with a grid interval of 1.0 Å. The outputs of our network are processed density slices of the same size.

### Data collection

In order to train and evaluate our EMReady framework, we have collected a non-redundant dataset of EM maps from the EMDB. All the single-particle EM entries at 3.0–6.0 Å resolutions that have associated PDB models are downloaded from the EMDB and PDB. Any EM map and its corresponding PDB structure that meet the following criteria are removed: (i) containing backbone atoms only, (ii) including unknown residues (UNK), (iii) including missing chain, (iv) having nonorthogonal map axis, and (v) resolution is not given by the FSC-0.143 cut-off. To ensure efficient training, we further exclude those entries with CC_mask values less than 0.75. The CC_mask values are calculated through the comparison between the deposited EM map and PDB model using phenix.map_model_cc. To remove the redundancy, the remaining cases are clustered using a greedy algorithm. Two models are considered to be similar if any chain in the first model has >30% sequence identity with any chain in the second model. The one with the largest number of similar cases is chosen as the

representative of the corresponding cluster, and then the cases in the cluster are removed. This procedure is repeated until all the cases are clustered. The final non-redundant set consists of the representatives of each cluster. A total of 436 pairs of EM maps and associated PDB structures with resolutions ranging from 3.0 Å to 6.0 Å are retained. Out of the total of 436 cases, 86 are randomly selected as the test set, 280 are randomly selected as the training set, and the remaining 70 maps are used as the validation set (Supplementary Data 11).

The initial test set consists of high-quality pairs of maps and PDB models with CC_mask values no less than 0.75. We then further collect a supplemental test set of entries with CC_mask values between 0.50 and 0.75. Greedy algorithm is also used to remove redundancy in the supplemental set using 30% as the sequence identity cut-off. More-over, we also exclude the cases in the supplemental set that have >30% sequence identity with any case in the above dataset of 436 cases. After adding 24 cases from the supplemental set, the final test set consists of 110 pairs of maps and structure models, as listed in Supplementary Data 1. As for half-maps, a subset of 25 pairs of half-maps is used, after excluding the cases in the test set that have no corresponding half-maps or have severe mismatch between the map and PDB structure (Supplementary Data 3). For individual chains, the density region within 4.0 Å of each protein or nucleic acid chain is segmented out of the whole primary map. Chains that have mismatch between atomic structure and density volume are excluded. The resulted set consists of 682 pairs of chains and density maps (Supplementary Data 5).

## Data preprocessing

During training, validation and testing, the grid size of experimental cryo-EM maps is unified to 1.0 Å by applying a cubic interpolation. The negative values for the map density are clipped at zero. The input density boxes of EMReady are of size $48 \times 48 \times 48$ and the output processed boxes are of the same size. In our previous work of EMNUSS[51], we normalized the density values in each input box to the range 0.0–1.0 by the maximum density value of each box. However, such local normalization is not suitable for the present task since it will introduce heterogeneity in the density amplitude for output maps. Thus, a global normalization strategy is adopted in the present study. Namely, we normalize the density values in each experimental map to the range 0–1.0 by the 99.999-percentile density value of each map. For each experimental EM density map in the training set, the target map is simulated from its associated PDB structure. For each experimental EM density map in the training set, the target density map is simulated from its associated PDB structure with a grid interval of 1.0 Å. Namely, given a PDB structure of $M$ atoms, the simulated density value $\rho$ on grid point $\mathbf{x}$ is calculated by the following formula

$$\rho(\mathbf{x}) = \sum_{i}^{M} \theta Z_i e^{-k|\mathbf{x}-\mathbf{r}_i|^2} \tag{1}$$

where $Z_i$ and $\mathbf{r}_i$ are the atomic number and the position vector of the $i$-th heavy atom ($i = 1, 2,..., M$), respectively. The value of $k$ depends on the reported resolution $R$ of the experimental map[52], i.e., $k = (\pi/(1.2 + 0.6R))^2$, and the scaling factor $\theta$ is defined as $\theta = (k/\pi)^{1.5}$.

Data augmentation is adopted in the training procedure. Specifically, the EM density maps and their corresponding simulated maps are first chunked into pairs of overlapping boxes of size $60 \times 60 \times 60$ with strides of 30 voxels. The inputs of training are augmented by random 90° rotations, and by randomly cropping a $48 \times 48 \times 48$ box from each $60 \times 60 \times 60$ box. To ensure effective training, non-positive boxes are excluded from training. For evaluation, the input EM density map is cut into overlapping boxes of size $48 \times 48 \times 48$ with strides of 12 voxels, which are then fed into the trained EMReady network. Finally, the output boxes are re-assembled into the final processed map by averaging the overlapping parts.

## Network training

The network is implemented through Pytorch1.8.1 + cuda11.1. Two different loss functions are adopted to calculate the difference between predicted volume slices and target slices. One is the smooth L1 loss, which calculates the local difference in the density values between predicted slices and target slices. The smooth L1 loss uses a squared term if the absolute element-wise error falls below 1.0 and an L1 term otherwise. The smooth L1 loss between a predicted slice $X$ and its corresponding target slice $Y$ is described by the following formula,

$$\text{SmoothL1Loss}(X,Y) = \sum_{i=1}^{N}\sum_{j=1}^{N}\sum_{k=1}^{N}\frac{l_{i,j,k}}{N^3} \tag{2}$$

where $N$ is the slice size ($N = 48$ in this study), and $l_{i,j,k}$ is the Smooth L1 distance between $X$ and $Y$ at position ($i, j, k$) described as follows,

$$l_{i,j,k} = \begin{cases} 0.5(X_{i,j,k} - Y_{i,j,k})^2, & \text{if } |X_{i,j,k} - Y_{i,j,k}|<1.0 \\ |X_{i,j,k} - Y_{i,j,k}| - 0.5, & \text{otherwise} \end{cases} \tag{3}$$

The other is SSIM loss which measures the non-local correlation between a predicted slice and its target slice according to their contrast and structure similarity. The contrast of a given slice is measured by its standard deviation of density values. Therefore, the contrast similarity $c(X, Y)$ of a pair of predicted slice and corresponding target slice can be described using the following equation,

$$c(X,Y) = \frac{2\sigma_X\sigma_Y}{\sigma_X^2 + \sigma_Y^2} \tag{4}$$

where $\sigma_X$ and $\sigma_Y$ are the standard deviations for the predicted slice $X$ and target slice $Y$, respectively. The structure similarity is the cosine similarity between two normalized slices as follows,

$$s(X,Y) = \left(\frac{1}{\sqrt{N^3-1}}\frac{X-\mu_X}{\sigma_X}\right) \times \left(\frac{1}{\sqrt{N^3-1}}\frac{Y-\mu_Y}{\sigma_Y}\right) = \frac{\sigma_{XY}}{\sigma_X\sigma_Y} \tag{5}$$

where $\mu_X$ and $\mu_Y$ are the mean density values for $X$ and $Y$, respectively, and $\sigma_{XY}$ is the covariance between $X$ and $Y$. Finally, the SSIM loss is simply given as follows,

$$\text{SSIMLoss}(X,Y) = 1 - c(X,Y) \times s(X,Y) = 1 - \frac{2\sigma_{XY} + \varepsilon}{\sigma_X^2 + \sigma_Y^2 + \varepsilon} \tag{6}$$

where $\varepsilon$ is set to be a small constant ($\varepsilon = 10^{-6}$ in this study) to prevent dividing by zero. We simply use the sum of Smooth L1 loss and SSIM loss as the total loss in the training. Adam optimizer is adopted to minimize the loss. Our networks are trained with 108 boxes employed in one batch. The initial learning rate is set to $5 \times 10^{-4}$, and will be reduced to 1/2 of its current value if the average loss on the training set does not decrease for every 4 continuous epochs. The training procedure will be stopped at 300 epochs, or when the learning rate reaches a minimum value of $1 \times 10^{-5}$. We have carefully considered various hyperparameters and different settings to optimize the performance of our EMReady method, including using a smaller batch size, and using regularization techniques like dropout and weight decay. However, as shown in the evaluation results in Supplementary Data 12 and 13 and in the training and validation loss curves in Supplementary Fig. 11 (Supplementary Data 14), compared to the baseline model, the models trained with other settings exhibit more or less underfitting. Besides, using a smaller batch size requires a drastically increased computation time to converge. The final choices of hyper-parameter used in baseline model were based on empirical observations and computational efficiency. The network model with the least loss on the validation set is used in the evaluation. During training, we use four NVIDIA A100 GPU cards of 40 GB VRAM, which can afford a

batch size of 108. During the evaluation, one A100 GPU card can afford a batch size of 180, but other GPU cards with at least 8 GB VRAM can also be used to run EMReady by reducing the batch size. As a comparison, the maximum batch size to run DeepEMhancer prediction on one A100 GPU card is about 51.

## Comparison with related methods

EMReady is compared with DeepEMhancer[38] and phenix.auto_sharpen[29] on the test set of 110 primary EM maps. It is worth mentioning that errors are reported by DeepEMhancer when processing 10 primary maps, and thus no results are given by DeepEMhancer on these maps. In terms of map interpretability, we compare EMReady with DeepEMhancer and phenix.auto_sharpen on 682 chains segmented from the test set of 110 primary maps. EMReady is compared with DeepEMhancer[38], LocScale[31], LocSpiral[35], phenix.auto_sharpen[29], and phenix.resolve_cryo_em (density modification)[33,34] on 25 pairs of half maps. It should be noted that there are three DeepEMhancer models: "tightTarget", "wideTarget" and "highRes". As different models are specialized for different situations, we report the combinatorial results of DeepEMhancer, where the DeepEMhancer "highRes" model is used on the maps with reported resolutions of <4 Å, and the default "tight-Target" model is used on the other cases, unless otherwise specified. The detailed results for all of the three DeepEMhancer models are listed in Supplementary Data. In addition, the masked maps we provide for LocScale are generated from the atomic structures using the same method of DeepEMhancer[38].

## Evaluations of EMReady

The performance of EMReady is exhaustively evaluated on the test sets of 110 primary maps, 25 pairs of half-maps, and 682 chains. The unmasked map-model Fourier shell correlation (FSC) is calculated for deposited and processed maps using phenix.mtriage[43]. The resolution at which the map-model FSC falls to one half (i.e., FSC-0.5) is used as a metric to measure the relative accuracy of the map[42]. Besides the FSC-0.5, we also use the MapQ plugin in UCSF Chimera[53] to measure the resolvability of density maps, i.e., Q-score[44]. Q-score measures the correlation between map density at each modeled atom and reference Gaussian density function. In addition, the correlation in real space can also be used to assess the quality of cryo-EM maps, which can be measured by the correlation coefficients (CC). Three CC values are calculated by phenix.map_model_cc[43] for a given pair of map and model: CC_box, CC_mask, and CC_peaks. The CC_box uses the entire map, the CC_mask uses the map values inside a mask calculated around the macromolecule, and the CC_peaks compares the map regions with the highest density values.

Furthermore, we also evaluated the role of EMReady in map interpretability through structural modeling. To avoid introducing the impact of human intervention, two automatic de novo model building tools, phenix.map_to_model[17–19] and MAINMAST[20], are used to build the structure models from the deposited maps and processed maps. The individual density patch for each chain is segmented out from the whole map using a distance cut-off of 4.0 Å. The entire set of 682 chains alone with their corresponding density patches are used in the evaluation of map interpretability for phenix.map_to_model. However, for MAINMAST that was designed to model full-length protein chains, only 385 protein chains without any gap in the PDB model are used in evaluation (Supplementary Data 6). The default parameters are provided for the main-chain tracing subprogram of MAINMAST. The density thresholds are optimally chosen for the deposited maps and processed maps. Specifically, a combination of threshold values (author recommended contour level × 0.25, 0.50, 0.75, and 1.00) is used for the deposited map. The same combination of density thresholds is also applied for the map processed by phenix.auto_sharpen, after shifting its density values to have equal mean and standard deviation values with the deposited map. The combinations

of density thresholds are chosen as [0.05, 0.10, 0.15, and 0.20] for DeepEMhancer-processed maps and as [0.5, 1.0, 1.5 and 2.0] for EMReady-processed maps. For each path generated by MAINMAST, 112 Cα models are generated with the parameter combinations described in the original paper of MAINMAST. The one with the highest threading score among all models is chosen as the final model for the given protein chain.

Two metrics calculated by phenix.chain_comparison are used to evaluate the accuracy of an atomic model built by phenix.map_to_model or MAINMAST: residue coverage and sequence match. The residue coverage is the fraction of the residues in one model matching the residues in another model within 3.0 Å regardless of their residue types. The Cα atom is used to represent the position of a residue for protein, and the P atom for nucleic acid. The sequence match is the percentage of the matched sequence with the same residue types in the target structure reproduced by the query model.

In addition, according to the Nyquist theorem, to obtain a structure model below 2.0 Å resolution, the cryo-EM reconstruction should have a grid size of 1.0 Å or smaller. With the improvement of new EM hardware and/or image processing technique, more and more high-resolution cryo-EM maps with a voxel size of <1.0 Å have been achieved, resulting in atomic-resolution protein structure determination[54]. If such maps are compulsively interpolated to a grid size of 1.0 Å, they will suffer from a loss in map information and/or quality. Therefore, we have also developed an EMReady model with a grid size of 0.5 Å to accommodate those maps with a voxel size of below 1.0 Å. Nevertheless, since cryo-EM maps at 3.0–6.0 Å resolutions often have a voxel size of above 1.0 Å, the primary 1.0 Å EMReady model is used as the default model in our evaluations, unless otherwise specified.

To conduct ablation experiments of EMReady, we trained four additional models including one EMReady model with input box size of 24 × 24 × 24, one EMReady model with input box size of 32 × 32 × 32, one EMReady model using the local UNet++ network[50], and one EMReady model without using SSIM loss. All the training hyperparameters (except for the changed one) were set to be same as those for the baseline EMReady model. Different from the baseline model, we used 4 as the window size of the swin transformer for the model of 32 × 32 × 32 input because the box size must be divisible by 8 times the window size.

## Calculation of local correlation map between maps

To assess the density modifications by EMReady, we evaluated the local correlation distribution of the deposited primary map and the EMReady-processed map of EMD-0257 relative to a simulated map that is generated from its associated PDB structure (PDB ID: 6HRA) using Eq. (1). The deposited map is interpolated to match the grid size of 1.0 Å of the EMReady-processed map and the simulated map. For each grid point with a positive density value on the deposited map or EMReady-processed map, we extracted a pair of small boxes with a size of 7 × 7 × 7 centered on that grid point from the map and the simulated map. We then calculated the Pearson's correlation between the two boxes to represent the local correlation at that point.

## Reporting summary

Further information on research design is available in the Nature Portfolio Reporting Summary linked to this article.

## Data availability

The data that support the findings of this study are available at [http://huanglab.phys.hust.edu.cn/EMReady/data/] or from the corresponding author upon request. The Source Data underlying Figs. 2–4c, d, 5–7, 8a, c, Tables 1, 2, Supplementary Figs. 2a, b, d, 4a–c, 6b, c, 7, 8a–d, 9, 11, and Supplementary Tables 1, 2 are provided as Source Data file. All published data sets used in this paper were taken from the EMDB

and PDB. A full list with links of the EMDB and PDB accession codes used in this study is available in Supplementary Data 15. Source data are provided with this paper.

## Code availability

The EMReady package and associated data are freely available for academic or non-commercial users at http://huanglab.phys.hust.edu.cn/EMReady/.

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

## Acknowledgements

This work was supported by the National Natural Science Foundation of China to S.H. (32161133002 and 62072199) and the startup grant of Huazhong University of Science and Technology.

## Author contributions

S.H. conceived and supervised the research. J.H. and S.H. designed the framework of EMReady. J.H. collected the dataset and implemented the code of EMReady. J.H. and T.L. conducted benchmarks. All authors analyzed the results. J.H. and S.H. wrote the manuscript. All authors read and approved the final version of the manuscript.

## Competing interests

The authors declare no competing interests.
