## [Peer Review File · Nature Communications]

Improvement of cryo-EM maps by simultaneous local and non-local deep learningReviewers' Comments:

Reviewer #1:

Remarks to the Author:

The article introduces a new method to enhance cryo-electron microscopy maps so they can be modelled at an atomic level. The method is based on deep learning. The method is technically sound and well explained (maybe the authors could explain the SwinTransformer, as it is not common knowledge, and the explanation would make the article self-contained).

The authors have extensively validated their method and compared their results to state-of-the-art algorithms. They show superior performance.

Overall, it is an excellent article with a very useful algorithm. This reviewer recommends publication of the article in its current form (maybe, with the explanation of the SwinTransformer requested above).

Reviewer #2:

Remarks to the Author:

The development of methods that aim at improving the interpretability of cryo-EM density maps remains an important challenge in view of the rapidly increasing number of cryo-EM structures that permit interpretation with atomic models. Raw cryo-EM are typically unsuited for atomic model building and are manipulated to facilitate interpretation by enhancing features at higher resolutions through filtering operations referred to as "post-processing". This paper describes a deep-learning-based post-processing method (EMReady) for cryo-EM maps by drawing on a training dataset of cryo-EM maps (input set) and a PDB-map set (target set).

Authors show improvements in terms of typical metrics as FSC-0.5, Q-scores and local correlations, when their approach is compared with typical sharpening methods as DeepEMhancer, phenix_autosharpen, and in some cases, LocSpiral and phenix.resolve_cryo_em. However, unfortunately I cannot recommend publication of this work in Nature Communications. Below are my general and specific comments on this work.

General Comments

1) I do see the proposed method as an incremental improvement when compared with other map postprocessing approaches, for example DeepEMHancer. While I believe that incremental improvements should always be welcomed, I think that this work lacks of the novelty required to be considered for publication in the Nature Communications journal, although I think it could be suitable for publication in another Nature's group journal after a careful review

2) I see this work as a threat to the Cryo-EM community. Typically, map post-processing is based on a sharpening process. Map sharpening is a term that collectively describes methods that aim at optimizing the resolution-dependent weighting of the amplitude component of Fourier coefficients in the Fourier representation of cryo-EM maps and thereby help (locally) improving clarity of high-resolution features in the real-space density map. This approach, however, would change completely (and without appropriate controls) the phases and amplitudes of experimental maps. Thus, what authors are proposing here is not anymore, a map sharpening approach, while it is something closer to a structure prediction approach based on deep learning. This approach is much riskier than any other map postprocessing approach, while I do not see here appropriate controls to evaluate cases where the method could overfit or hallucinate.

3) Following my previous comment, one of the main differences of this approach with DeepEMHancer is the use of PBD maps as targets, instead of LocScaled maps. PDB maps are more different than

LocScaled maps to input maps, thus, EMReady approach will require a more complex model with higher number of parameters to learn the mapping. It is known that the more complex the proposed model is, the higher the computational demands are, and also the risk to produce overfitting when generalizing. Thus, predicting PDB maps directly from mid-resolution maps would indeed achieving "resolution improvements", something that cannot be obtained without important biases and the risk of producing maps affected by overfitting or artefacts. Moreover, I see that PDB maps show homogeneous quality, thus, all regions show the same quality and resolution. On the other hand, experimental EM maps typically show heterogeneous quality and resolution. This summed to the soft L1 loss used by EMReady points me even more to the capacity of this model to overfit. Note that EMReady will essentially be rewarded to focus more on map regions that are very different to the PDB map (and thus require "creative" solutions) to optimize the network parameters, while easier but safer patches will have less importance. For me these increases more the risk of overfitting and hallucinations since it is encouraging the network to modify more aggressively the parts of the map with lower quality.

Thus, I propose the following actions:

3.1) First you should provide more details about the used architecture, number of trainable parameters, memory requirements during prediction and during training. Does your model fit in a typical GPU card?

3.2) You must provide the training curves for the loss for the training set and the validation/evaluation set for the different epochs.

3.3) You should do additional control experiments to show that the network is not overfitting or causing hallucinations. To this end, I propose the following ones:

- Obtain the prediction of the network for a map showing very high pure noise (Gaussian white noise should be OK) after being preprocessed (normalized) by your approach. (max at 1 approximately and min at 0 approximately).

- For the evaluation set, show the regions producing the highest differences between the predicted map and the PDB map.

- Obtain the prediction for a very low SNR EMDB map not included in your training set with a resolution of 5 A or worse, obtained by a low number of particles ~50K particles and without (or not too much) symmetry so the SNR should be low. This will require to reconstruct a high-resolution example with a smaller subset of particles to study the robustness against noise.

- Obtain the prediction of 3DEM maps that are likely incorrect as the well known EMD-5447.

For all these cases, evaluate the results obtained and analyze the behaviour of the model and its capacity to exhibit overfitting and/or hallucinations.

4) The main advance of this approach may be summarized by the following sentence:

"During the training of EMReady, we use simulated cryo-EM maps from associated structures from the Protein Data Bank (PDB) instead of optimized experimental cryo-EM maps as the target maps, which can avoid the noise and/or error issue of experimental maps in DeepEMhancer"

First, it is false that PDB atomic models are perfect, and they usually show errors. According to [1], 31% of all models examined in their analysis possess unrealistic occupancies or/and B-factor values, such as all being set to zero or other unlikely values. They also reported that 40% of models analysed

show cross-correlations between cryo-EM maps and respective models below 0.5, which is very poor. Thus, assuming that PDBs are perfect will lead to biases in the analysis.

[1] Afonine, P. V. et al. New tools for the analysis and validation of cryo-EM maps and atomic models. *Acta Crystallogr. D Struct. Biol.* 74, 814–840 (2018).

5) Moreover, and following my previous comment, regions that are not modelled in the PDBs will never be learnt from the model and will be automatically masked out or will be reconstructed erroneously. This points me directly to the effect that parts of the density typically not modelled in PDBs will be filtered out in the EMReady maps. I think it would be important to also mention the pitfalls of such effects.

As one example, glycan chains are essential and often functionally important in many proteins, but because of their flexibility often they appear blurred in cryo-EM reconstructions and therefore, frequently not modelled. A lack of such density in EMReady-generated training dataset will therefore likely lead to bias in the learned model and may result in filtering out such densities in the EMReady maps. If true, this could affect accuracy and completeness of such maps in these important map features. The above is of course not restricted to glycan structures, but could be any posttranslational modification, lipids, ligands and even some protein motives. that may not be sufficiently covered in the training set. I consider it important to show such examples if the authors have identified them, or to specifically investigate them if they have not, for demonstrating whether or not this represents a point of caution or concern.

6) Network architecture. Swing transformers usually are not totally non-local and has a region of influence, which is surely not the whole map because memory constrains. Indeed, since the authors process the Map in chunks of 48x48x48 voxels, claiming that they are considering non-local features is a bit misleading. I think that what the authors can claim at most is that their architecture follows state-of-the-art computer vision designs that try to capture local and non-local relationship within a limited distance. In addition to that, which are the memory requirements of this network? Compared to DeepEMhancer?

7) Why the improvements reported are about? Are they because the use of PDB maps instead of LocScaled maps? Are about the loss employed? Are about the use of Swing transformers? I see answering this question relevant. I propose to retrain the DeepEMHancer model against your training dataset to check if the most important aspect is the use of PDB maps and conduct an ablation study to determine the most important features of the newly proposed architecture

More specific Comments

Introduction

8) I do not agree with the following sentence in the introduction section:

8.1- "... local sharpening approaches relies heavily on the use of some a priori information about the target map like masks to distinguish the macromolecule from the noise, an estimation of the local resolution of the map, or the structural information of atomic models."

Can you be more specific about the heavy prior information requirements of the target maps needed? Do you consider a coarse solvent mask and/or a local resolution map estimation a heavy use of a priori information? In addition to this, do you have systematic tests to show the dependency of these approaches over these parameters?

8.2- "Another critical issue is that cryo-EM maps often exhibit a long-range non-local heterogeneity over the entire map due to the conformational and compositional in imaged structures and the

preferred orientations in single particles^{3, 37}, which poses a major challenge to traditional residual convolutional networks that focus on local modeling ability.

What does it mean “long-range non-local heterogeneity” and “global heterogeneity”? Please clarify these terms. In addition, I do not see why this is an issue. If the model was trained with maps showing the same resolution (or similar) the network will perfectly know how to map these patches. Why is this a “critical” issue? Having said that, the claim that their method is better suited to deal with “long-range non-local heterogeneity” seems arbitrary and I would suggest the users to remove it if they cannot show any illustrative example of how their method helps in such case.

Map-model Fourier shell correlation

9) The first comment that I have is about the use of masked map-model FSC-0.5 resolutions (FSC-0.5) for the test set of 110 deposited primary cryo-EM maps. I suppose that authors run the different approaches and finally mask the resulting maps by the very same mask. Otherwise, results will be wrong. Moreover, how these masks are created? Are binary or soft masks? In addition to these, authors should also recalculate the FSC-0.5 for the different raw postprocessing maps without any masking operation since automatic masking is not a solved problem

10) Secondly, I do not agree with the terminology used here as it seems that the different postprocessing approaches have the capacity to improve the resolution of reconstructions. This is not true. The resolution should be obtained from half maps computed from angular refinement methods after gold-standard approach but not by map postprocessing methods. Agree with using the FSC-0.5 metric to evaluate the similarities between the prediction and target volumes and check if the model is learning but it should be crystal clear that postprocessing approaches are useful for visualization purposes only and that resolution improvement should never be claimed

11) -“On average, EMReady achieved an FSC-0.5 resolution of 3.64 Å, which is significantly improved from 4.42 Å for the deposited maps, while DeepEMhancer and phenix.auto sharpen yielded little improvement on the deposited primary maps.”

This sentence is very weird to me as in the DeepEMHancer paper authors clearly show that DeepEMHancer can improve the map-model FSC-0.5 when compared to deposited maps. Even more surprising, I understand that the intersection between the EMReady evaluation set and the DeepEMhancer training set should be large. Thus, this result should be even higher. Can you clarify this discrepancy? The authors should also report the version they used for the different cases, and the model type they employed. In addition, to make the comparison fairer, DeepEMhancer highRes model should be used when executed on high resolution maps, whereas the default model should be used for the other cases.

Atom resolvability

12) -Following my previous comment, authors should also obtain the Q scores for raw postprocessing maps not further masked.

Model-map correlation

13) -CC_box, CC_mask and CC_peaks are not defined at any place in the manuscript.

14) Again, and following my previous comment in point 11, the results shown in Figures 2 and 3 are very weird to me considering the results shown in DeepEMhancer’s paper and my own experience. I have to say that I do not believe these figures, especially the comparisons between deposited and DeepEMhancer maps. Please include in all these figures the results from deposited maps (in panels b,

d and f are not included) and the results obtained by LocScale approach. These results also should be obtained from raw postprocessing maps (without additional masking). The authors should probably upload the post-processed maps to the public domain so that we could assess the differences if the obtained results disagree with the results expected according to the literature.

Examples of improved maps

15) "However, EMReady effectively improves the contrast of the region between the macromolecule and the lipid nanodisc, making the lipid region almost invisible in the processed map"

I really want to see this raw postprocessed map (without any additional masking) at a very low level where the noise could be seen clearly. Is this noise structured? Do exist hallucinations and the network identify noise patterns as protein structure (loops, etc)?

16) Why the FSCs shown in Figure 4 do not go to zero at Nyquist frequency?

Improvement on half-maps

17) I do not understand why LocSpiral and phenix.resolve_cryo_em are not evaluated over final EM maps and only over half-maps. What I see more worrying is why in the results of Table 3, LocSpiral and phenix.resolve_cryo_em are not taken into consideration. In addition to that why in the figures 5 and 6 density modification is taken into consideration and but not in the corresponding tables?

18) To me is not clear how are obtained the results shown in Table 2 and Figures 5 and 6. Are half-maps postprocessed and then averaged to a final postprocessed map, which is then compared to the PDB map? or are half-maps postprocessed and then compared by the FSC-0.5? This should be clearly presented. How are the Q-scores and CC's obtained from the half-maps?

19) " It can be seen from Table 2 that only EMReady significantly improved the masked map-model FSC-0.5 resolutions of the half-maps among the five post-processing methods (Fig. 5a)."

As commented before, the FSC-0.5 should be also obtained from raw postprocessing maps without any additional masking operation. Please include also unmasked map-model FSC-0.5 resolutions.

Improvement in map interpretability

20) "Therefore, we further evaluate the performance of EMReady in terms of de novo model building on a test set of 682 chains of cryo-EM maps"

Are these chains from the evaluation set only? Indeed, it would to be better to analyze data not included in the training neither in the evaluation sets (note that here there is no validation and testing set).

Evaluations against higher-resolution structures

21) Do exist maps of the apoferritin in the training set? If this is the case, the results shown here could be overoptimistic.

Network architecture:

22) Very little information is given about the architecture used. No background information about what a Swing transformer is and why it is different from typical convolutions. Which is the influence region of these transformers? I guess that developing a network that could estimate true global correlations

may be too expensive. What is the number of parameters required by this architecture and the memory requirements for training and evaluation?

Data preprocessing:

23) "For training, the EM density maps and their corresponding simulated maps are chunked into pairs of overlapping boxes of size $60 \times 60 \times 60$ with strides of 30 voxels. To ensure effective training, nonpositive boxes are excluded from training. For evaluation, the EM density map is cut into overlapping boxes of size $48 \times 48 \times 48$ with strides of 12 voxels, which are then fed into the trained EMReady network. Finally, the output boxes are re-assembled into the final processed map by averaging the overlapping parts."

Why using a box of $60 \times 60 \times 60$ with strides of 30 voxels? Have you tried different sizes and evaluated the performance? I think it would be good to see such an analysis. Why the boxes are different for training than for evaluation?

Network training:

24) Why the loss training curves for the training and evaluation sets are now shown? Please include these figures.

25) As commented above, the usage of PDB maps as target together with the soft L1 loss proposed here clearly favours the learning from regions that are much different between EM and PDB maps. Considering that the quality of an EM map may vary a lot, while PDB maps show homogeneous quality, I see that it may exist regions that are very different between these maps, while others may be very similar. Thus, the proposed loss will only allow the learning from a low number of different (problematic) regions. This issue clearly points me to overfitting. This comment added over the presumable large network used (the authors do not provide the number of learnable parameters) makes me very concerned about the capacity of the model to overfit. Additionally, Adam optimizer is well known for its capacity to overfit. Moreover, the authors surprisingly do not provide the learning curves over the training and validation sets for the different epochs.

Data availability:

26) I do think necessary to publish all data supporting the findings of this work, including the training and evaluation sets with corresponding PDB and experimental maps (not only the accessory codes)

Code availability

27) Is EMReady close to non-academic research groups? Have the authors plans to protect this method via a patent? If so, it must be explicitly said in the competing interest sections.

Manuscript ID: NCOMMS-22-31369-T

Title: Improvement of cryo-EM maps by simultaneous local and non-local deep learning

Author(s): Jiahua He; et al.

We very much appreciate the valuable comments/suggestions from the reviewers. We have conducted necessary computations/analyses and revised our manuscript accordingly. The revised parts in the manuscript are highlighted in red. The point-to-point responses to the comments are listed as follows.

Reviewer #1 (Remarks to the Author):

The article introduces a new method to enhance cryo-electron microscopy maps so they can be modelled at an atomic level. The method is based on deep learning. The method is technically sound and well explained (maybe the authors could explain the SwinTransformer, as it is not common knowledge, and the explanation would make the article self-contained).

The authors have extensively validated their method and compared their results to state-of-the-art algorithms. They show superior performance.

Overall, it is an excellent article with a very useful algorithm. This reviewer recommends publication of the article in its current form (maybe, with the explanation of the SwinTransformer requested above).

Response: We thank the reviewer for reviewing our manuscript and giving the positive comments and valuable suggestions. We have added the detailed explanation about the swin (shifted window) transformer in our revised manuscript. [Page 4, last paragraph]

**

Reviewer #2 (Remarks to the Author):

The development of methods that aim at improving the interpretability of cryo-EM density maps remains an important challenge in view of the rapidly increasing number of cryo-EM structures that permit interpretation with atomic models. Raw cryo-EM are typically unsuited for atomic model building and are manipulated to facilitate interpretation by enhancing features at higher resolutions through filtering operations referred to as “post-processing”. This paper describes a deep-learning-based post-processing method (EMReady) for cryo-EM maps by drawing on a training dataset of cryo-EM maps (input set) and a PDB-map set (target set).

Authors show improvements in terms of typical metrics as FSC-0.5, Q-scores and local correlations, when their approach is compared with typical sharpening methods as DeepEMhancer, phenix_autosharpen, and in some cases, LocSpiral and phenix.resolve_cryo_em. However, unfortunately I cannot recommend publication of this work in Nature Communications. Below are my general and specific comments on this work.

Response: We thank the reviewer for reviewing our manuscript and giving the valuable comments and suggestions. We have conducted extensive computations/investigations and revised our manuscript accordingly.

General Comments

1) I do see the proposed method as an incremental improvement when compared with other map postprocessing approaches, for example DeepEMHancer. While I believe that incremental improvements should always be welcomed, I think that this work lacks of the novelty required to be considered for publication in the Nature Communications journal, although I think it could be suitable for publication in another Nature's group journal after a careful review.

Response: We thank the reviewer for the comments. We guess that the reviewer may somehow miss some important points of our EMReady method. In our opinion, EMReady represents a major advancement of post-processing approaches for Cryo-EM maps. Compared with existing approaches, the novelty of EMReady lies in not only the use of simulated maps, but also the local and non-local modeling capability in the SCUNet network as well as the combination of local smooth L1 loss and non-local SSIM loss in the training process. To avoid these points being missed, we have added additional discussions in the corresponding places of our manuscript. [Page 4, last paragraph; Page 5, 1st paragraph; Page 17, last paragraph; Page 20, 1st paragraph]

2) I see this work as a threat to the Cryo-EM community. Typically, map post-processing is based on a sharpening process. Map sharpening is a term that collectively describes methods that aim at optimizing the resolution-dependent weighting of the amplitude component of Fourier coefficients in the Fourier representation of cryo-EM maps and thereby help (locally) improving clarity of high-resolution features in the real-space density map. This approach, however, would change completely (and without appropriate controls) the phases and amplitudes of experimental maps. Thus, what authors are proposing here is not anymore, a map sharpening approach, while it is something closer to a structure prediction approach based on deep learning. This approach is much riskier than any other map postprocessing approach, while I do not see here appropriate controls to evaluate cases where the method could overfit or hallucinate.

Response: We thank the reviewer for raising the concern. Actually, we do see EMReady as a valuable addition to the cryo-EM community instead of a threat to the Cryo-EM community. As we mentioned in the Introduction, post-processing methods may be roughly

divided into two groups. One is the traditional amplitude-reweighting-based map sharpening methods, and the other is the deep learning-based map post-processing approaches. The two groups of methods have their own advantages and disadvantages. For examples, the traditional map sharpening methods maintain the original phase and amplitudes, but tend to adjust the map density moderately. In contrast, the deep learning-based methods tend to modify the density more intensively according to learned confidence, but will change the original phase and amplitudes. We think that both groups of methods should not be a threat to each other, but a valuable complementarity of each other. In addition, we have conducted extensive validation of the density modification by EMReady, which clearly show that our EMReady method does not overfit or hallucinate. [Pages 14-17, Section 2.6]

Moreover, given the preferences of different post-processing methods, we have added some discussions about the use of EMReady as "...despite the superior performance of EMReady, users should not solely rely on the EMReady-processed map when building a model from a cryo-EM map. They are strongly recommended to try different post-processing methods and examine their modeling results by orthogonal ways.". [Page 21, last paragraph]

3) Following my previous comment, one of the main differences of this approach with DeepEMHancer is the use of PBD maps as targets, instead of LocScaled maps. PBD maps are more different than LocScaled maps to input maps, thus, EMReady approach will require a more complex model with higher number of parameters to learn the mapping. It is known that the more complex the proposed model is, the higher the computational demands are, and also the risk to produce overfitting when generalizing. Thus, predicting PBD maps directly from mid-resolution maps would indeed achieving "resolution improvements", something that cannot be obtained without important biases and the risk of producing maps affected by overfitting or artefacts. Moreover, I see that PBD maps show homogeneous quality, thus, all regions show the same quality and resolution. On the other hand, experimental EM maps typically show heterogeneous quality and resolution. This summed to the soft L1 loss used by EMReady points me even more to the capacity of this model to overfit. Note that EMReady will essentially be rewarded to focus more on map regions that are very different to the PBD map (and thus require "creative" solutions) to optimize the network parameters, while easier but safer patches will have less importance. For me these increases more the risk of overfitting and hallucinations since it is encouraging the network to modify more aggressively the parts of the map with lower quality.

Response: We thank the reviewer for the comments. We guess that the reviewer may have miss some important points of EMReady. As we mentioned before, the main differences of our EMReady approach with DeepEMHancer are not only the use of PBD maps as targets, but also the use of the local and non-local modeling capability in the SCUNet network as well as the combination of local smooth L1 loss and non-local SSIM loss in the training process. To avoid these points being missed, we have added additional discussions in the corresponding places of our manuscript. [Page 4, last paragraph; Page 5, 1st paragraph; Page 17, last paragraph; Page 20, 1st paragraph]

Regarding the network, despite the higher powerfulness of EMReady than DeepEMhancer, our EMReady network only have 4717952 trainable parameters, which are much fewer than the 51119889 trainable parameters for DeepEMhancer. [Supplementary Data 8]

During the development of EMReady, we have used three independent data sets, including training, validation, and testing sets. The model with the least validation loss is selected to evaluate on the independent test set, so as to avoid overfitting. It is indeed the fact that using smooth L1 loss alone to train the network is in risk of overfitting. However, we actually use the combined loss of smooth L1 loss and SSIM loss, where the latter term has been proved to be effective in reducing possible overfitting by the former term. If the deep learning model is overfitted to the mapping from experimental map to the simulated map of homogeneous quality, it will give similar good results to maps of different resolutions, give a homogeneous result to a map of heterogeneous quality and give mistakenly modified results to maps not seen in the training set. We did not see such evidences of overfitting or hallucination in our independent test sets and in the additional control experiments that we added following the reviewer's suggestion. [Pages 14-17, Section 2.6]

We have added a separate section to validate the density modification by EMReady, which clearly show that our EMReady method does not overfit or hallucinate. In fact, EMReady tends to confidently modify those higher-resolution or rigid regions, while cautiously do limited modifications to those lower-resolution or flexible regions, so as to maintain the original density signals while improving the quality of the map, which is contrary to the reviewer's above comment of "For me these increases more the risk of overfitting and hallucinations since it is encouraging the network to modify more aggressively the parts of the map with lower quality." [Pages 14-17, Section 2.6; Page 15, 2nd paragraph]

Thus, I propose the following actions:

3.1) First you should provide more details about the used architecture, number of trainable parameters, memory requirements during prediction and during training. Does your model fit in a typical GPU card?

Response: We thank the reviewer for raising these questions. We have added the details of our network architecture in Supplementary Data 8. We have also added the details about number of trainable parameters and VRAM requirements during training and prediction in the revised manuscript. Specifically, EMReady has 4717952 trainable parameters. During the training, four NVIDIA A100 GPU cards of 40GB VRAM can afford a batch size of 108. During the evaluation, Our EMReady model can fit to any typical GPU cards as long as they have a minimum memory of 8 G and support CUDA of version 11.1 or later. [Page 22, 1st paragraph; Page 26, 1st paragraph; Supplementary Data 8]

3.2) You must provide the training curves for the loss for the training set and the validation/evaluation set for the different epochs.

Response: We have provided the training curves of EMReady in our revised manuscript. [Supplementary Fig. 4a,b; Supplementary Data 7]

3.3) You should do additional control experiments to show that the network is not overfitting or causing hallucinations. To this end, I propose the following ones:

- Obtain the prediction of the network for a map showing very high pure noise (Gaussian white noise should be OK) after being preprocessed (normalized) by your approach. (max at 1 approximately and min at 0 approximately).

Response: We thank the reviewer for the suggestion. We have analyzed the impact of noise on a simulated map with very high pure noise. It was shown that EMReady successfully suppressed the noises and recovered the density volume of the simulated map from the noisy map. [Page 16, last paragraph; Page 17, 1st paragraph; Supplementary Fig. 3a-c]

-For the evaluation set, show the regions producing the highest differences between the predicted map and the PDB map.

Response: We thank the reviewer for the suggestion. We have conducted local correlation analysis of the predicted map and the simulated PDB map on target EMD-0257. It is shown that “the highest density difference tends to occur in the low resolution/flexible regions instead of the higher resolution/rigid regions. Similar trend can also be observed in the density difference between the deposited map and the simulated map. These results suggest that EMReady does not simply modify the deposited map towards the simulated one, but is intelligent enough to detect the local quality of the deposited map and then modifies the density based on a reliable way. [Page 15, last paragraph; Page 29, Section 4.7; Supplementary Fig. 2d]

-Obtain the prediction for a very low SNR EMDB map not included in your training set with a resolution of 5 Å or worse, obtained by a low number of particles ~50K particles and without (or not too much) symmetry so the SNR should be low. This will require to reconstruct a high-resolution example with a smaller subset of particles to study the robustness against noise.

Response: We thank the reviewer for the suggestion. We have analyzed the result of EMD-2678 that was reconstructed from 37310 particles at 5.4 Å resolution. Compared to the deposited map, the map processed by EMReady was de-noised and showed higher map-map FSC over very long range of inverse resolutions against a higher-resolution reference, EMD-3061 at 3.4 Å resolution. [Page 13, last paragraph; Page 14, 2nd paragraph; Supplementary Fig. 1c,d]

-Obtain the prediction of 3DEM maps that are likely incorrect as the well known EMD-5447.

Response: We thanked the reviewer for the suggestion. We have investigated the EMReady-processed map of EMD-5447. It was shown that EMReady did very few modifications to this map. [Page 16, 2nd paragraph; Supplementary Fig. 2e]

For all these cases, evaluate the results obtained and analyze the behaviour of the model and its capacity to exhibit overfitting and/or hallucinations.

Response: We have evaluated the results obtained by EMReady and analyzed the behavior of the EMReady model for all those cases mentioned by the reviewer. It was shown that our EMReady model did not exhibit overfitting and/or hallucinations. [See the responses above]

4) The main advance of this approach may be summarized by the following sentence:

“During the training of EMReady, we use simulated cryo-EM maps from associated structures from the Protein Data Bank (PDB) instead of optimized experimental cryo-EM maps as the target maps, which can avoid the noise and/or error issue of experimental maps in DeepEMhancer”

First, it is false that PDB atomic models are perfect, and they usually show errors. According to [1], 31% of all models examined in their analysis possess unrealistic occupancies or/and B-factor values, such as all being set to zero or other unlikely values. They also reported that 40% of models analysed show cross-correlations between cryo-EM maps and respective models below 0.5, which is very poor. Thus, assuming that PDBs are perfect will lead to biases in the analysis.

[1] Afonine, P. V. et al. New tools for the analysis and validation of cryo-EM maps and atomic models. *Acta Crystallogr. D Struct. Biol.* 74, 814–840 (2018).

Response: We thank the reviewer for the comments. We guess that the reviewer may have miss some important points of EMReady. As we mentioned before, the main differences of our EMReady approach with DeepEMhancer are not only the use of PBD maps as targets, but also the use of the local and non-local modeling capability in the SCUNet network as well as the combination of local smooth L1 loss and non-local SSIM loss in the training process. To avoid these points being missed, we have added additional discussions in the corresponding places of our manuscript. [Page 4, last paragraph; Page 5, 1st paragraph; Page 17, last paragraph; Page 20, 1st paragraph]

We agree that the PDB structures are not perfect and “40% of models analyzed show cross-correlations between cryo-EM maps and respective models below 0.5”. However, we have tried to minimized the possible biases introduced by the PDB structures by using strict criteria of model-map correlation quality when collecting the training cases, as described in Section 4.2 (Data collection). There did not exist any significant evidence suggesting that the PDB models used in our study lead to significant biases in the analysis. [Page 22, last

paragraph]

5) Moreover, and following my previous comment, regions that are not modelled in the PDBs will never be learnt from the model and will be automatically masked out or will be reconstructed erroneously. This points me directly to the effect that parts of the density typically not modelled in PDBs will be filtered out in the EMReady maps. I think it would be important to also mention the pitfalls of such effects.

As one example, glycan chains are essential and often functionally important in many proteins, but because of their flexibility often they appear blurred in cryo-EM reconstructions and therefore, frequently not modelled. A lack of such density in EMReady-generated training dataset will therefore likely lead to bias in the learned model and may result in filtering out such densities in the EMReady maps. If true, this could affect accuracy and completeness of such maps in these important map features. The above is of course not restricted to glycan structures, but could be any posttranslational modification, lipids, ligands and even some protein motives. that may not be sufficiently covered in the training set. I consider it important to show such examples if the authors have identified them, or to specifically investigate them if they have not, for demonstrating whether or not this represents a point of caution or concern.

Response: We thank the reviewer for pointing out the important issues and valuable suggestions. We have added the corresponding parts to discuss about the limitations with EMReady due to the effects of some unmodelled regions and special components like ligands, glycans, post-translational modifications, lipids, and protein motives. [Page 21, 1st paragraph]

6) Network architecture. Swing transformers usually are not totally non-local and has a region of influence, which is surely not the whole map because memory constrains. Indeed, since the authors process the Map in chunks of 48x48x48 voxels, claiming that they are considering non-local features is a bit misleading. I think that what the authors can claim at most is that their architecture follows state-of-the-art computer vision designs that try to capture local and non-local relationship within a limited distance. In addition to that, which are the memory requirements of this network? Compared to DeepEMhancer?

Response: We thank the reviewer for the nice comments and suggestions. We have added a separate paragraph to discuss about this issue. We have also added the details about memory requirements of the network. Specifically, during the training, four NVIDIA A100 GPU cards of 40GB VRAM can afford a batch size of 108. During evaluation, one A100 GPU card can afford a batch size of 180, but other GPU cards with at least 8G VRAM can also be used to run EMReady by reducing the batch size. As a comparison, DeepEMhancer requires more memory to run, and the maximum batch size to run DeepEMhancer prediction on one A100 card is about 51. [Page 20, 2nd paragraph; Page 26, 1st paragraph]

7) Why the improvements reported are about? Are they because the use of PDB maps instead of LocScaled maps? Are about the loss employed? Are about the use of Swing transformers?

I see answering this question relevant. I propose to retrain the DeepEMHancer model against your training dataset to check if the most important aspect is the use of PDB maps and conduct an ablation study to determine the most important features of the newly proposed architecture.

Response: We thank the reviewer for raising the issue and giving the valuable suggestions. We have conducted ablation experiments of the nonlocal components of our EMReady framework and added a separate section to discuss about the ablation results. It was revealed that the most important feature of the newly proposed architecture was the swin Transformer in the network, followed by the SSIM loss and then the input box size. [Pages 17-19, Section 2.7; Page 28, last paragraph, Supplementary Table 2]

However, the direct ablation experiment of the simulated PDB maps for EMReady is difficult because EMReady cannot take the LocScale-processed maps that have been used by DeepEMHancer as the training set, because LocScale is designed for unfiltered and unsharpened maps only. For the similar reason, DeepEMHancer also cannot be trained on the training set of primary maps that have been used by EMReady, because DeepEMHancer is designed to use half-maps for training. Instead, we have adopted an alternative way to analyze and discuss the impact of using the simulated PDB maps on the EMReady model. [Page 19, 2nd paragraph]

More specific Comments

Introduction

8) I do not agree with the following sentence in the introduction section:

8.1- "... local sharpening approaches relies heavily on the use of some a priori information about the target map like masks to distinguish the macromolecule from the noise, an estimation of the local resolution of the map, or the structural information of atomic models."

Can you be more specific about the heavy prior information requirements of the target maps needed? Do you consider a coarse solvent mask and/or a local resolution map estimation a heavy use of a priori information? In addition to this, do you have systematic tests to show the dependency of these approaches over these parameters?

Response: We thank the reviewer for the nice comments. To avoid misunderstanding, we have modified the sentence to "... and some local sharpening approaches rely on the use of a priori information about the target map ..." in the revised manuscript. [On the top of Page 4]

8.2- "Another critical issue is that cryo-EM maps often exhibit a long-range non-local heterogeneity over the entire map due to the conformational and compositional in imaged structures and the preferred orientations in single particles^{3, 37}, which poses a major

challenge to traditional residual convolutional networks that focus on local modeling ability.

What does it mean “long-range non-local heterogeneity” and “global heterogeneity”? Please clarify these terms. In addition, I do not see why this is an issue. If the model was trained with maps showing the same resolution (or similar) the network will perfectly know how to map these patches. Why is this a “critical” issue? Having said that, the claim that their method is better suited to deal with “long-range non-local heterogeneity” seems arbitrary and I would suggest the users to remove it if they cannot show any illustrative example of how their method helps in such case.

Response: We thank the reviewer for the nice comments. To avoid misunderstanding, we have removed the statement from our manuscript. [Page 4, 2nd paragraph]

Map-model Fourier shell correlation

9) The first comment that I have is about the use of masked map-model FSC-0.5 resolutions (FSC-0.5) for the test set of 110 deposited primary cryo-EM maps. I suppose that authors run the different approaches and finally mask the resulting maps by the very same mask. Otherwise, results will be wrong. Moreover, how these masks are created? Are binary or soft masks? In addition to these, authors should also recalculate the FSC-0.5 for the different raw postprocessing maps without any masking operation since automatic masking is not a solved problem.

Response: We thank the reviewer for the comments. The mask used in calculating the masked map-model FSC-0.5 for a given map is a soft mask calculated automatically by phenix.mtriage from the corresponding PDB model [Acta Crystallogr. D Struct. Biol. 2018, 74, 814–840]. The same mask is applied to the deposited map and maps processed by different methods to ensure that the comparison of FSC-0.5 is meaningful. As the masked map-model FSC is expected to be a better measure of the quality of the map around a model [Nat Methods. 2020, 17, 923-927], we focus on the masked map-model FSC-0.5 in this study. However, following the reviewer’s request, we also report the unmasked map-model FSC-0.5 (Supplementary Table 1), where the significantly better performance of EMReady than the other methods can also be observed. [Page 6, last paragraph; Page 10, 1st paragraph; Page 27, 1st paragraph; Supplementary Table 1]

10) Secondly, I do not agree with the terminology used here as it seems that the different postprocessing approaches have the capacity to improve the resolution of reconstructions. This is not true. The resolution should be obtained from half maps computed from angular refinement methods after gold-standard approach but not by map postprocessing methods. Agree with using the FSC-0.5 metric to evaluate the similarities between the prediction and target volumes and check if the model is learning but it should be crystal clear that postprocessing approaches are useful for visualization purposes only and that resolution improvement should never be claimed.

Response: We thank the reviewer for the nice comment. We have revised our manuscript to avoid using the terminology of “improve resolution”. [The entire manuscript]

11) -“On average, EMReady achieved an FSC-0.5 resolution of 3.64 Å, which is significantly improved from 4.42 Å for the deposited maps, while DeepEMhancer and phenix.auto sharpen yielded little improvement on the deposited primary maps.”

This sentence is very weird to me as in the DeepEMHancer paper authors clearly show that DeepEMHancer can improve the map-model FSC-0.5 when compared to deposited maps. Even more surprising, I understand that the intersection between the EMReady evaluation set and the DeepEMhancer training set should be large. Thus, this result should be even higher. Can you clarify this discrepancy? The authors should also report the version they used for the different cases, and the model type they employed. In addition, to make the comparison fairer, DeepEMhancer highRes model should be used when executed on high resolution maps, whereas the default model should be used for the other cases.

Response: We thank the reviewer for the comments. The poor evaluation results of map-model FSC-0.5 by DeepEMhancer come from two sources. On one hand, the input maps are primary maps that usually have been masked/sharpened by their authors, thus are hard for DeepEMhancer to further improve. On the other hand, the FSC-0.5 reported in Fig. 2a,b is actually the masked FSC-0.5. We have also reported the unmasked FSC in the revised manuscript. As shown in Supplementary Table 1, DeepEMhancer can indeed improve the unmasked FSC-0.5. [Page 6, last paragraph; Page 10, 1st paragraph; Page 27, 1st paragraph; Supplementary Table 1]

The version of DeepEMhancer we used is 0.14. Following the reviewer’s suggestions, we have used the combinatorial result of DeepEMhancer in our revised manuscript. That is, the results of DeepEMhancer “highRes” model are used on the maps with reported resolutions of <4 Å and the results of the default “tightTarget” model are used otherwise. We also have reported the results of three different DeepEMhancer models (“tightTarget”, “wideTarget” and “highRes”) in the Supplementary Data. [Page 26, last paragraph; Pages 34-35, Tables 1, 2; Pages 38-43, Figs. 2, 3, 5, 6, 7; Supplementary Data 1, 3, 4]

Atom resolvability

12) -Following my previous comment, authors should also obtain the Q scores for raw postprocessing maps not further masked.

Response: We thank the reviewer for raising the issue. In fact, Q-scores were not masked. We only used mask in FSC calculations. We have further clarified this in the manuscript. [Page 27, 1st paragraph]

Model-map correlation

13) -CC_box, CC_mask and CC_peaks are not defined at any place in the manuscript.

Response: We thank the reviewer for pointing out the issue. We have added the descriptions of CC_box, CC_mask, and CC_peaks in our revised manuscript. [Page 27, 1st paragraph]

14) Again, and following my previous comment in point 11, the results shown in Figures 2 and 3 are very weird to me considering the results shown in DeepEMhancer's paper and my own experience. I have to say that I do not believe these figures, especially the comparisons between deposited and DeepEMhancer maps. Please include in all these figures the results from deposited maps (in panels b, d and f are not included) and the results obtained by LocScale approach. These results also should be obtained from raw postprocessing maps (without additional masking). The authors should probably upload the post-processed maps to the public domain so that we could assess the differences if the obtained results disagree with the results expected according to the literature.

Response: We thank the reviewer for raising the concerns. As we mentioned before, the discrepancy of DeepEMhancer's results between DeepEMhancer's paper and this manuscript is because the FSC-0.5 reported in Figures 2 and 3 is actually the masked FSC-0.5. We have also reported the unmasked FSC in the revised manuscript. As shown in Supplementary Table 1, DeepEMhancer can indeed improve the unmasked FSC-0.5. [Page 6, last paragraph; Page 10, 1st paragraph; Supplementary Table 1]

Actually, Fig. 2b,d, Fig. 3b,d,f, and Fig. 5b,d already contain the results of the deposited map (the results of deposited primary maps or half-maps are represented by the coordinates of the x-axis, and the results of the processed maps of different methods are represented by the coordinates of the y-axis). In addition, following the reviewer's suggestion, we have applied LocScale to the test set of half-maps. [Page 26, last paragraph; Page 35, Table 2; Pages 41-42, Figs. 5, 6]

We have also published all the input and post-processed maps of DeepEMhancer to our website (<http://huanglab.phys.hust.edu.cn/EMReady/>). [Page 29, Data availability]

Examples of improved maps

15) "However, EMReady effectively improves the contrast of the region between the macromolecule and the lipid nanodisc, making the lipid region almost invisible in the processed map"

I really want to see this raw postprocessed map (without any additional masking) at a very low level where the noise could be seen clearly. Is this noise structured? Do exist hallucinations and the network identify noise patterns as protein structure (loops, etc)?

Response: Following the reviewer's suggestion, we have displayed the raw EMReady-processed map at a very low level in Supplementary Fig. 3d. It can be seen from the figure

that the noise is not structured. There does not exist hallucination. [Page 17, at the bottom of 1st paragraph; Supplementary Fig. 3d]

16) Why the FSCs shown in Figure 4 do not go to zero at Nyquist frequency?

Response: We thank the reviewer for raising the question. Nyquist frequency defines the resolution limit that 3D reconstruction can reach, at which the signal of a given 3D reconstruction cannot be distinguished from noise due to the limited sampling rate. Therefore, the FSC between two independent half-reconstructions (with the same signal but different noises) would go to zero at Nyquist frequency. However, things are different for map-model FSC. More generally, the map-map FSC of half-reconstructions indicate the true resolution (by FSC-0.143), but the map-model FSC (or FSC_{ref}) is not the resolution but an FSC-based resolution-dependent estimate of map similarity to an ideal reference map (Terwilliger TC, Ludtke SJ, Read RJ, Adams PD, Afonine PV. Improvement of cryo-EM maps by density modification. Nat Methods. 2020;17(9):923-927. doi:10.1038/s41592-020-0914-9). There is no noise in such reference model map. Therefore, when the noise component of a give map is suppressed, it will yield a high map-model FSC curve. The figure below shows the map-model FSC curves of the DeepEMhancer-processed map and the simulated map with 1.0 Å sampling rate drawn onto Fig. 4d. It can be seen that the DeepEMhancer-processed map also gives a positive FSC at Nyquist, and that the clean simulated map has even higher FSC at Nyquist frequency.

Improvement on half-maps

17) I do not understand why LocSpiral and phenix.resolve_cryo_em are not evaluated over final EM maps and only over half-maps. What I see more worrying is why in the results of Table 3, LocSpiral and phenix.resolve_cryo_em are not taken into consideration. In addition to that why in the figures 5 and 6 density modification is taken into consideration and but not in the corresponding tables?

Response: We thank the reviewer for the comments. It is because these methods only accept two half-maps as the input.

(please see https://phenix-online.org/documentation/reference/resolve_cryo_em.html and <https://cosmic-cryoem.org/tools/locspiral/>).

The test set of 682 chains were taken from the primary maps, so we did not evaluate LocSpiral nor phenix.resolve_cryo_em on this test set. In addition, density modification is actually phenix.resolve_cryo_em, which is already in the figures 5 and 6. We have modified the labels in Figs. 5, 6 to avoid misunderstanding. [Pages 41-42, Figs. 5, 6]

18) To me is not clear how are obtained the results shown in Table 2 and Figures 5 and 6. Are half-maps postprocessed and then averaged to a final postprocessed map, which is then compared to the PDB map? or are half-maps postprocessed and then compared by the FSC-0.5? This should be clearly presented. How are the Q-scores and CC's obtained from the half-maps?

Response: We thank the reviewer for pointing out this issue. The input of EMReady is actually the average map of two half-maps. The evaluations metrics of half-maps are also based on such averaged maps. We have clarified this issue in our revised manuscript. [Page 10, 1st paragraph]

19) “ It can be seen from Table 2 that only EMReady significantly improved the masked map-model FSC-0.5 resolutions of the half-maps among the five post-processing methods (Fig. 5a).”

As commented before, the FSC-0.5 should be also obtained from raw postprocessing maps without any additional masking operation. Please include also unmasked map-model FSC-0.5 resolutions.

Response: We thank the reviewer for the comments. Following the reviewer's suggestion, we have provided all the unmasked map-model FSC-0.5 values and discussed about the results accordingly in the revised manuscript. [Page 6, last paragraph; Page 10, 1st paragraph; Page 27, 1st paragraph; Supplementary Table 1]

Improvement in map interpretability

20) “Therefore, we further evaluate the performance of EMReady in terms of de novo model building on a test set of 682 chains of cryo-EM maps”

Are these chains from the evaluation set only? Indeed, it would be better to analyze data not included in the training neither in the evaluation sets (note that here there is no validation and testing set).

Response: We thank the reviewer for raising the questions. We apologize for causing the confusion. In fact, we have divided our data sets into independent training, validation and

test sets for EMReady. Therefore, the 682 chains we analyzed here are taken from the test set, which are not included in the training neither in the validation sets. We have clarified this issue in our revised manuscript. The detailed list of training and validation sets can be found in Supplementary Data 9. [Page 5, last paragraph; Page 23, 1st and 2nd paragraphs; Supplementary Data 9]

Evaluations against higher-resolution structures

21) Do exist maps of the apoferritin in the training set? If this is the case, the results shown here could be overoptimistic.

Response: We thank the reviewer for raising the concern. The maps of the apoferritin do not exist in the training set.

Network architecture:

22) Very little information is given about the architecture used. No background information about what a Swin transformer is and why it is different from typical convolutions. Which is the influence region of these transformers? I guess that developing a network that could estimate true global correlations may be too expensive. What is the number of parameters required by this architecture and the memory requirements for training and evaluation?

Response: We have provided the background information of swin transformer in the revised manuscript. The swin transformer is an efficient transformer that combines self-attention of non-overlapping local windows and non-local cross-window connection by shifted window partitioning. The network of EMReady has 4717952 trainable parameters. During training, four NVIDIA A100 GPU cards of 40GB VRAM can afford a batch size of 108. During evaluation, one such GPU card can afford a batch size of 180, but other GPU cards can also be used to run EMReady as long as they have a minimum memory of 8G. [Page 4, last paragraph; Page 22, Section 4.1; Page 26, 1st paragraph]

Data preprocessing:

23) “For training, the EM density maps and their corresponding simulated maps are chunked into pairs of overlapping boxes of size $60 \times 60 \times 60$ with strides of 30 voxels. To ensure effective training, nonpositive boxes are excluded from training. For evaluation, the EM density map is cut into overlapping boxes of size $48 \times 48 \times 48$ with strides of 12 voxels, which are then fed into the trained EMReady network. Finally, the output boxes are re-assembled into the final processed map by averaging the overlapping parts.”

Why using a box of $60 \times 60 \times 60$ with strides of 30 voxels? Have you tried different sizes and evaluated the performance? I think it would be good to see such an analysis. Why the boxes are different for training that for evaluation?

Response: We thank the reviewer for pointing out the issue. We apologize for causing the confusion. Actually, we are using $48 \times 48 \times 48$ as the input box size in both training and evaluation. The larger box size of $60 \times 60 \times 60$ is merely used in data augmentation, that is, the inputs of training are augmented by randomly cropping a $48 \times 48 \times 48$ box from each $60 \times 60 \times 60$ box. We have clarified this issue in our revised manuscript. [Page 24, 2nd paragraph]

We have also evaluated the performance of EMReady using two different box sizes in the ablation experiments of our revised manuscript. [Pages 17-19, Section 2.7; Supplementary Table 2]

Network training:

24) Why the loss training curves for the training and evaluation sets are now shown? Please include these figures.

Response: We have provided the learning curves on the training and validation sets for the different epochs in the revised manuscript. [Supplementary Fig. 4a,b; Supplementary Data 7]

25) As commented above, the usage of PDB maps as target together with the soft L1 loss proposed here clearly favours the learning from regions that are much different between EM and PDB maps. Considering that the quality of an EM map may vary a lot, while PDB maps show homogeneous quality, I see that it may exist regions that are very different between these maps, while others may be very similar. Thus, the proposed loss will only allow the learning from a low number of different (problematic) regions. This issue clearly points me to overfitting. This comment added over the presumable large network used (the authors do not provide the number of learnable parameters) makes me very concerned about the capacity of the model to overfit. Additionally, Adam optimizer is well known for its capacity to overfit. Moreover, the authors surprisingly do not provide the learning curves over the training and validation sets for the different epochs.

Response: We thank the reviewer for the comments. In fact, we do not simply use smooth L1 loss in EMReady, but use the combinatorial loss function of smooth L1 loss and SSIM loss. We have demonstrated that incorporating SSIM loss into the training process effectively prevents EMReady from possible overfitting introduced by smooth L1 loss. [Pages 17-19, Section 2.7; Supplementary Table 2; Supplementary Fig. 4]

We have added a separate section to validate the density modification by EMReady on additional control experiments, which clearly show that there does not exist any significant evidence of overfitting in our trained model. [Pages 14-17, Section 2.6; Supplementary Figs. 1-3]

We have provided the learning curves over the training and validation sets for the different epochs in the revised manuscript. [Supplementary Fig. 4a,b; Supplementary Data 7]

Data availability:

26) I do think necessary to publish all data supporting the findings of this work, including the training and evaluation sets with corresponding PDB and experimental maps (not only the accessory codes)

Response: Following the reviewer's suggestion, we have published all data supporting the findings of this work, including the training and evaluation sets with corresponding PDB and experimental maps, in <http://huanglab.phys.hust.edu.cn/EMReady/>. [Page 29, Data availability]

Code availability

27) Is EMReady close to non-academic research groups? Have the authors plans to protect this method via a patent? If so, it must be explicitly said in the competing interest sections.

Response: The authors declare no competing interests. EMReady is released under GNU General Public License Version 3. [Page 29, Code availability]

Reviewers' Comments:

Reviewer #1:

Remarks to the Author:

The authors have performed a substantial improvement of their article. From my point of view, it is ready to be published.

Reviewer #2:

Remarks to the Author:

This is my second revision of this manuscript. While I can appreciate the work done by the authors from the previous work, I regret to say that I still see major points that worries me significantly. Obviously, I can be wrong or over pessimistic, but I think that it is important to discuss them further. Moreover, taking into account that the manuscript has had only one technical revision (the other reviewer only had minor comments), I believe that is necessary to consider the feedback from an independent third reviewer that can evaluate the work at the light of my previous comments. My major concern is presented in points 7 and 8 of the author responses, in which the authors include valuable data, but only as a minor supplementary data that is barely mentioned in the main text. This is important as they directly point to their results supporting their method. Authors should present unmasked FSC plots (and resolution estimations) as their main results and not masked FSCs (or both together). Unmasked FSC plots can provide misleading results, if they are not considered jointly with the non masked FSCs. These results should appear in the main text and not hidden in Supplementary material sections. The discussions should be done according to unmasked FSC analysis not masked ones. Additionally, authors have to clearly explain the discrepancy between the results obtained by DeepEMhancer in their results and the ones provided in the literature. Another important point is that authors should tone down their claims about overfitting, and non-local learning capacity of the network (actually limited to regions of 48x48x48. Below I provide my specific comments:

1) To show that EMReady does not overfit (in the cryo-EM sense, that is, do not generate artifacts from noise) and hallucinate, authors conducted different tests. At the end of page 13, authors show masked map-map FSCs between EMD-20028 against EMD-20026 and EMD-2677 against EMD-3061 before and after applying EMReady. As I said in my previous review, FSCs between masked maps are extremely easy to cheat by playing with the mask. Thus, showing unmasked FSCs is required to assess the quality of maps. Moreover, in Supplementary Fig. 1 (b-c), it is shown that EMD-2678 (5.4 Å resolution) show higher FSC than EMD-2677 (4.5 Å) at high resolutions, which directly contradict the statement: "Comparing the above three cases also reveals that the map improvement by EMReady is resolution-dependent. The higher the resolution of a map is, the more the improvement is (Supplementary Fig. 1a-c)." Thus, it is not necessarily true that "Such trend can be understood because the density information is more reliable for a map with a higher resolution and thus EMReady is more confident to modify its densities".

2) Authors display in Supplementary Fig 2 a figure to show that EMReady does not overfit. This test is interesting but to me could be biased. Note that the local correlation is low only in regions with local resolution of $\sim 7\text{Å}$ or worse. I see this behaviour absolutely normal as EMReady has been trained with maps showing global resolutions between 3-6Å, and only in regions with an associated PDB model. I do not see very likely or frequent to have very low-resolution map regions with an associated traced PDB model in EMDB. I guess that EMReady does not know what to do when it is exposed to these regions, as was not trained with similar data, and simply does nothing. Contrary, if we look in Supplementary Fig 2 (b) to regions with resolutions between 4-6Å, (resolutions included in the training set) these regions provide high correlations in Fig. 2 (d). Moreover, the upper part of the protein has low resolution and not low correlations. Therefore, to this reviewer this example is not a clear example to show that EMReady does not have the capacity to overfit.

The same comment applies to the test done just below for EMD-5447 at the beginning of page 16

(Supplementary Fig. 2), which is aimed to show that EMReady does not “simply modify the deposited map towards the simulated one but is intelligent enough to detect the local quality of the deposited map and then modifies the density based on a reliable way”. I do not see these tests fully convincing for the reasons explained above.

Moreover, I do not see the window sizes used to calculate local correlations. I think authors must clearly show these window sizes to calculate local correlations. Did the authors change their sizes along the different tests?

Finally, I see it interesting to show the deposited and EMReady processed maps for EMD-0257 and EMD-5447 to evaluate these maps visually.

3) The authors should show visual examples of the improvement obtained by EMReady when compared with other postprocessing approaches. They could show different secondary structures at different resolutions (4Å, 3Å, 2Å, for example) showing the improvements provided by EMReady in the visualization of side-chains (zoom-in images).

4) About the tests done to evaluate the robustness of the method to noise, there is something that I do not understand. In Supplementary Fig 3 a-c authors show the capacity of EMReady to denoise simulated maps. What I do not understand here is why the method has the capacity to remove noise as in the training phase it learnt only how to map input maps to PDBs, and PDBs have not noise. According to this example, when EMReady sees patterns not seen before during training it removes them. On the other hand, in Supplementary Fig 3 d authors show that when EMReady finds the lipid nanodisk it does not remove it or maps it to a higher resolution structure like the ones in macromolecules (I have also some comments about this, which appear below). The same behaviour happens when EMReady finds low resolution structures, which were not seen during training. How can this have happened?

5) I do see that authors have not done one of my previous requirements, which I do see important: “Obtain the prediction of the network for a map showing very high pure noise (Gaussian white noise should be OK) after being preprocessed (normalized) by your approach. (max at 1 approximately and min at 0 approximately).” Please include this experiment, the one introduced is noise+simulated protein not pure noise.

6) About the plots of loss vs epochs, Supplementary figure 4 is difficult to understand because it shows the loss vs. the architecture + loss type. So, the Y-axis value depends on the configuration they evaluate. They should add at least two other plots that are L1 vs epochs, SSIM vs epochs for each of the tested architectures. Then we could better compare the effect of loss type, since in your tables (Supplementary Data 5 and 6) it is difficult to appreciate any effect, $\sim 0.05 - 0.07\text{Å}$ difference is significant? For me it does not seem to be. In these tables you must report a measure of dispersion, such as the standard deviation. Better than these sup tables would be much better to show boxplots or violinplots.

7) Along the revised paper (and in this response to reviewer document) authors have mentioned many times the following idea about the major advancements of EMReady: “the local and non-local modeling capability in the SCUNet network as well as the combination of local smooth L1 loss and non-local SSIM loss in the training process.” In the response to referee 6) Network architecture, they include the following paragraph in the paper: “Despite the powerfulness of non-local modeling in EMReady, it should be noted that the nonlocal range cannot be global but have a limited distance because of the high memory cost of swin transformer during the training of EMReady. Namely, the window size of swin transformer and the size of input density box, which affect the non-local modeling range of EMReady, is strongly limited by the total memory of GPUs during the training, though the running of the trained EMReady model is not memory expensive and can be accommodated on a GPU with a memory as low as 8G. It is expected that the non-local modeling capability of EMReady can be further improved with more advanced GPUs with higher memory available.”

As I mentioned in my previous comment, “the authors process the map in chunks of 48x48x48 voxels, claiming that they are considering non-local features is a bit misleading” Thus, the non-local capacities of EMReady is restricted to regions of 48x48x48 voxels, which is quite local. Authors should indicate this limitation (with the actual numbers) explicitly in the paper and relax most of the sentences about the capacity of EMReady to capture local and non-local information and about the novelty/major advances of their approach and in practice it is not the case. The authors are free to claim that they use newer architectures and network components, but they shouldn’t claim that non-locality is the main reason for the better performance.

8) Map-model Fourier shell correlation. I appreciate the effort done by authors including the 0.5 FSC resolutions for unmasked maps in supplementary tables. However, as I mentioned before, FSCs are extremely sensitive to used masks. Additionally, reducing the FSC to a single number is a huge lost of information. Thus, I see extremely important to show along the different figures of the paper (not only in supplementary material) always plots of unmasked FSCs. If authors want to show masked FSCs as well, it is OK, but always the unmasked plots should be shown to avoid misleading conclusions.

9) Weird results of DeepEMHancer. First, I see this response a very important example of why including unmasked FSC results in the main manuscript is absolutely mandatory. Masked FSC results are biased by the solvent mask used and could provide misleading results. Please include ALL unmasked results in the manuscript and NOT ONLY in supplementary tables and figures. I do see this point totally mandatory. Additionally, I do not understand the explanation provided by the authors. When you mention ‘deposited primary map’, I (and likely everyone in the field) would think that this is the refined map before any postprocessing step. Additionally, the deposited map FSC resolution is the one provided from half maps without any postprocessing step. But according to your comment, this is an actual postprocessed map and because of this “the input maps are primary maps that usually have been masked/sharpened by their authors, thus are hard for DeepEMhancer to further improve”. To me this does not take any sense. If the primary map deposited in EMDB is postprocessed, then you should take the average from half maps. I continue thinking that authors should clearly explain the discrepancies they obtained between the DeepEMhancer results expected according to the literature and what they obtained in their results and that this should be included in the main text. Second thing is directly asking to authors, if in their test sets they have explicitly looked for deposited maps, which were processed by DeepEMhancer in their publications. In this case, they results may be biased and their conclusions compromised. Can you please provide the percentage of maps in the test set that were processed by DeepEMhancer in their publications?

As these comparisons are your primary results supporting your method, I do see this discussion very worrying. Also, I am concerned about authors providing only masked FSC results in the main text and main figures, and only some raw information in supplementary material without any discussion. The only thing authors said about these unmasked results is: “In addition to the masked map-model FSC, we also report the unmasked map-model FSC-0.5 (Supplementary Table 1), where the significantly better performance of EMReady than other methods can also be observed.”

“We have also published all the input and post-processed maps of DeepEMhancer to our website (<http://huanglab.phys.hust.edu.cn/EMReady/>). [Page 29, Data availability]”

Sorry, I cannot find these results in the website.

10) Previous referee comment (15)

“Following the reviewer’s suggestion, we have displayed the raw EMReadyprocessed map at a very low level in Supplementary Fig. 3d. It can be seen from the figure 12 that the noise is not structured. There does not exist hallucination. [Page 17, at the bottom of 1st paragraph; Supplementary Fig. 3d]”

To my eye, Supplementary Fig. 3, show alpha helix like structures in the lipid region. Although the helix-like structure resemblance could be argued, what it is undeniable from this figure is that the noise is always not structured.

11) Previous referee comment (16). I do not see the answer providing by authors satisfying. Both map-to-map and map-to-model FSCs should go to zero at Nyquist. This is a typical sanity check. Authors should investigate why FSCs obtained from EMReady processed maps does not go to zero when compared to atomic models.

Reviewer #3:

Remarks to the Author:

The authors present a cryo-EM map post-processing approach, called EMReady. It is based on a 3D convolutional neural network that is trained on pairs of experimental cryo-EM maps as input and simulated data as ground truth. They present a series of tests for evaluating the performance of the model, primarily in terms of improved atom resolvability and map interpretability, and results that support improvements in some of these aspects over other methods, like DeepEMhancer. Although the approach is novel and has great potential, the evaluation is lacking and there are several unsubstantiated claims made by the authors.

MAJOR ISSUE 1: Map interpretability using phenix.map to model

Phenix.map to model is no longer state-of-the-art in automated model building. Additionally, it is reliant on a series of map segmentations that should particularly benefit from data-driven segmentation preprocessing. There are several deep-learning based methods that perform automated model building today. This includes Deept racer, MAINMAST and ModelAngelo. The authors are strongly encouraged to evaluate the sequence coverage and/or recall using one or two of these methods after processing the map using EMReady. If map quality is truly improved (beyond just improved segmentation or chain separation), this should be reflected in the results from these methods as well.

MAJOR ISSUE 2: Atom resolvability using Q-score and model-to-map FSC

The Q-score measures how well the experimental maps fit with Gaussian-like functions around identified atoms. The authors present a neural network that is trained on simulated ground truth data that is generated using a Gaussian forward model, see equation (1). Hence, the L1-loss used in the training of the model is indirectly also maximizing the Q-score. Naturally, the output of any processing using this network will exhibit more Gaussian-like densities, which will inflate the Q-score. Subsequently, there is little to no value in using the Q-score to evaluate the performance of the method as this particular metric was indirectly optimized throughout training. The same applies also to model-to-map FSC, as the model is first converted to a gaussian mixture model before the FSC is calculated.

MAJOR ISSUE 3: Map quality

Although the map becomes perhaps more interpretable to humans there are no results indicating improvements to the quality of the reconstruction itself through for instance denoising or local sharpening/filtering. The generated maps increase similarity to a simple gaussian forward model, which might not be the true underlying signal that is observed in cryo-EM. This is however neither evaluated or clarified by the authors. The authors could evaluate this by for instance applying EMReady to one of the half-maps and calculate the FSC between the output and the other unprocessed half-map. Notice that the FSC_{0.5} between two processed half-maps will be inflated due to injection of the same signal/bias into both half-maps, which is similar to the solvent mask effect, see (Chen Ultramicroscopy 2013).

MAJOR ISSUE 4: Method applicability and overall evaluation

Although the authors mention at the end of the discussion section that “users should not solely rely on the EMReady-processed map when building a model from a cryo-EM map”, more results need to be presented that evaluate when the approach is expected to fail. For instance, what are the outliers in Figure 7a and what actually happens to ligands or lipids?

MINOR ISSUES

Why is the validation and testing dataset so large relative to the training dataset?

Why are the volume chunks “48x48x48”? How does the performance change with chunk-size? The batch-size is unusually large. Have the authors tried smaller batch-sizes? How does this affect convergence and overfitting? What about dropout and weight decay?

The word “chunk” is used extensively throughout the results and method section to refer to volume slices. The word slice, patch or section are more suited alternatives.

The use of the word “primary map” is used to refer to final reconstruction results as opposed to half-maps. This needs to be explicitly clarified as this phrase lacks a general meaning within the field.

Manuscript ID: NCOMMS-22-31369A

Title: Improvement of cryo-EM maps by simultaneous local and non-local deep learning

Author(s): Jiahua He; et al.

We very much appreciate the valuable comments/suggestions from the three reviewers. We have conducted necessary computations/analyses and revised our manuscript accordingly. The revised parts in the manuscript are highlighted **in red**. The point-to-point responses to the comments are listed as follows.

Reviewer #1 (Remarks to the Author):

The authors have performed a substantial improvement of their article. From my point of view, it is ready to be published.

Response: We thank Reviewer #1 for reviewing our manuscript and giving the positive comment.

Reviewer #2 (Remarks to the Author):

This is my second revision of this manuscript. While I can appreciate the work done by the authors from the previous work, I regret to say that I still see major points that worries me significantly. Obviously, I can be wrong or over pessimistic, but I think that it is important to discuss them further. Moreover, taking into account that the manuscript has had only one technical revision (the other reviewer only had minor comments), I believe that is necessary to consider the feedback from an independent third reviewer that can evaluate the work at the light of my previous comments.

My major concern is presented in points 7 and 8 of the author responses, in which the authors include valuable data, but only as a minor supplementary data that is barely mentioned in the main test. This is import as they directly point to their results supporting their method. Authors should present unmasked FSC plots (and resolution estimations) as their main results and not masked FSCs (or both together). Unmasked FSC plots can provide misleading results, if they are not considered jointly with the non masked FSCs. These results should appear in the main text and not hidden in Supplementary material sections. The discussions should be done according to unmasked FSC analysis not masked ones. Additionally, authors have to clearly explain the discrepancy between the results obtained by DeepEMhancer in their results and the ones provided in the literature. Another important point is that authors should tone down their claims about overfitting, and non-local learning capacity of the network (actually

limited to regions of 48x48x48. Below I provide my specific comments:

Response: Thank you for your thoughtful and thorough review and giving the valuable comments/suggestions. We appreciate your continued engagement and the opportunity to address your concerns. We understand the importance of careful evaluation and welcome the additional feedback from an independent third reviewer. We have carefully considered your comments regarding the evaluations and claims made in the manuscript. We have revised the manuscript to address these concerns and provide more comprehensive evaluations of our approach. We hope that the revised manuscript can address your concern and meet your expectations.

1) To show that EMReady does not overfit (in the cryo-EM sense, that is, do not generate artifacts from noise) and hallucinate, authors conducted different tests. At the end of page 13, authors show masked map-map FSCs between EMD-20028 against EMD-20026 and EMD-2677 against EMD-3061 before and after applying EMReady. As I said in my previous review, FSCs between masked maps are extremely easy to cheat by playing with the mask. Thus, showing unmasked FSCs is required to assess the quality of maps. Moreover, in Supplementary Fig. 1 (b-c), it is shown that EMD-2678 (5.4 Å resolution) show higher FSC than EMD-2677 (4.5 Å) at high resolutions, which directly contradict the statement: “Comparing the above three cases also reveals that the map improvement by EMReady is resolution-dependent. The higher the resolution of a map is, the more the improvement is (Supplementary Fig. 1a-c).” Thus, it is not necessarily true that “Such trend can be understood because the density information is more reliable for a map with a higher resolution and thus EMReady is more confident to modify its densities”.

Response: Following the reviewer’s suggestion, we have replaced all the masked FSC results with the unmasked ones in the main manuscript and figures. We ensure that unmasked FSC plots are always shown along different figures in our revised manuscript and that our discussions are based on unmasked FSCs.

[Supplementary Fig. 3; The entire manuscript]

We apologize for the confusion caused by the use of different x-ranges in our original figure, and have reprepared the figure with the same range of 0-0.4 Å⁻¹ for the x-axis. As shown in the figure below, the masked and unmasked FSCs of EMD-2678 are actually lower than EMD-2677 at high resolutions before and after applying EMReady. Nevertheless, to avoid possible overgeneralizing of the conclusions, we have revised our claims regarding the resolution-dependency of EMReady. Specifically, we now state in the revised manuscript that “Comparing the above three cases also reveals that the map improvement by EMReady may to some extent depend on the resolution.” and “Such trend can be partially attributed to the fact that EMReady tends to be more confident to modify the densities in a map with higher resolution and reliable density information”.

[Page 15, last paragraph; Supplementary Fig. 3]

2) Authors display in Supplementary Fig 2 a figure to show that EMReady does not overfit. This test is interesting but to me could be biased. Note that the local correlation is low only in regions with local resolution of $\sim 7\text{\AA}$ or worse. I see this behaviour absolutely normal as EMReady has been trained with maps showing global resolutions between 3-6 \AA , and only in regions with an associated PDB model. I do not see very likely or frequent to have very low-resolution map regions with an associated traced PDB model in EMD. I guess that EMReady does not know what to do when it is exposed to these regions, as was not trained with similar data, and simply does nothing. Contrary, if we look in Supplementary Fig 2 (b) to regions with resolutions between 4-6 \AA , (resolutions included in the training set) these regions provide high correlations in Fig. 2 (d). Moreover, the upper part of the protein has low resolution and not low correlations. Therefore, to this reviewer this example is not a clear example to show that EMReady does not have the capacity to overfit.

The same comment applies to the test done just below for EMD-5447 at the beginning of page 16 (Supplementary Fig. 2), which is aimed to show that EMReady does not “simply modify the deposited map towards the simulated one but is intelligent enough to detect the local quality of the deposited map and then modifies the density based on a reliable way”. I do not see these tests fully convincing for the reasons explained above. Moreover, I do not see the window sizes used to calculate local correlations. I think authors must clearly show these window sizes to calculate local correlations. Did the authors change their sizes along the different tests?

Finally, I see interesting to show the deposited and EMReady processed maps for EMD-0257 and EMD-5447 to evaluate these maps visually.

Response: We thank the reviewer for raising the issue. Actually, this figure was originally used to address the reviewer’s previous suggestion of “For the evaluation set, show the regions producing the highest differences between the predicted map and the PDB map” and to investigate the impact of local resolution/quality on the density modification of EMReady. For better illustration, we have reprepared and reorganized the panels of the figure.

We agree that the test may have limitations and that the behavior observed in regions with local resolutions of $\sim 7\text{\AA}$ or worse is expected, given that EMReady is trained on

the maps with global resolutions between 3-6 Å and only in regions with associated PDB models. We agree that these two examples may not provide a conclusive demonstration of EMReady's ability to detect local quality and modify density according to the local quality. To avoid overgeneralizing the conclusions based on the examples, we have toned down our claims about overfitting and removed the statements about local quality-dependency in the revised manuscript. However, we still believe that our experiment on these two test cases that are independent from the training set at least provides a useful indication that EMReady is not only robust to improve the regions with resolutions similar to those included in the training set, but also would not likely to create artifacts for lower-resolution and/or lower-quality regions. Thank you again for your helpful feedback, and we hope that our revision addresses your concern. [Page 16, last paragraph; Page 17, 1st paragraph; Supplementary Fig. 4]

Regarding the abnormally low local resolution on the top of EMD-0257, we found that it is due to the biased local resolution estimation using a single primary map. We have replaced the single map with the pair of half-maps in the local resolution estimation in our revised manuscript. Now, the local resolution distribution is consistent with the B-factor distribution in the reference PDB structure.

[Supplementary Fig. 4a]

Regarding the window size to calculate local correlations, we apologize for any confusion caused. The window size used to calculate local correlations is $7 \times 7 \times 7$, and this information can be found in Section 4.7 of the manuscript. We did not change the window size during different tests. Thank you for bringing this to our attention.

[Page 31, Section 4.7]

Following the reviewer's suggestion, we have shown the comparison between deposited and EMReady-processed maps for EMD-0257 (Supplementary Fig. 4b,c) and EMD-5447 (Supplementary Fig. 4e) in our revised manuscript.

[Supplementary Fig. 4]

3) The authors should show visual examples of the improvement obtained by EMReady when compared with other postprocessing approaches. They could show different secondary structures at different resolutions (4Å, 3Å, 2Å, for example) showing the improvements provided by EMReady in the visualization of side-chains (zoom-in images).

Response: Following the reviewer's suggestion, we have included the visual examples that demonstrate the improvements obtained by EMReady when compared with other post-processing approaches. We have shown the examples of different secondary structures at different resolutions as zoom-in images in Supplementary Fig. 1 and discussed them accordingly in the revised manuscript.

[Page 9, at the bottom of 2nd paragraph; Supplementary Fig. 1]

4) About the tests done to evaluate the robustness of the method to noise, there is something that I do not understand. In Supplementary Fig 3 a-c authors show the capacity of EMReady to denoise simulated maps. What I do not understand here is why the method has the capacity to remove noise as in the training phase it learnt only how to map input maps to PDBs, and PDBs have not noise. According to this example, when EMReady sees patterns not been seen before during training it removes them. On the other hand, in Supplementary Fig 3 d authors show that when EMReady finds the lipid nanodisk it does not remove it or maps it to a higher resolution structure like the ones in macromolecules (I have also some comments about this, which appear below). The same behaviour happens when EMReady finds low resolution structures, which were not seen during training. How can this have happened?

Response: We thank the reviewer for raising the concern. The reason why EMReady has the capacity to denoise maps is that during the training phase, the network learned how to represent and extract meaningful features from the input maps, including the noise. It should be emphasized that there are noises in the training experimental maps. Therefore, by training on pairs of experimental cryo-EM maps as input and simulated data as ground truth, EMReady learned how to suppress noises in maps.

It is worth mentioning that network was trained on a variety of structures, including those with lipid nanodiscs (e.g. EMD-8911, EMD-9000, EMD-21132, etc). We have to clarify that EMReady does not “remove” the noises. It just suppresses the noises by improving the contrast between the signals of the actual structure and those of the other components in the density map. This functionality of EMReady stems from the loss function, that is, by suppressing the densities of non-structural components, the losses can also be reduced in the training procedure, but in a safer manner compared to removing them. That’s why EMReady just suppresses but not removes the lipid nanodisc in Supplementary Fig. 5d. Another proof to this is shown in Supplementary Fig. 5e, where the gaussian noises also exist in the EMReady-processed map, but only at extremely low contour threshold.

I understand the question raised by the reviewer regarding the issue of EMReady not removing the low-resolution or mapping them to higher-resolution structures. On one hand, lower-resolution structures at least to some extent share similar features with higher-resolution structures, which can be recognized by the network. Since these lower-resolution structures lack higher-resolution patterns that can be recognized and improved by EMReady, they tend to remain unmodified. On the other hand, it is also worth noting that the network architecture of EMReady includes skip connections, which allows the information from earlier layers to be retained and combined with the information from later layers. This feature can help the network to preserve the information from lower-resolution structures and incorporate it into the final prediction, even if these structures were not explicitly seen during training. It is expected that for these lower-resolution/lower-quality structures that was not seen during training, the raw density of the input map would dominate the prediction through skip connection.

We hope that our explanations can address the reviewer's concern.

5) I do see that authors have not done one of my previous requirements, which I do see important: "Obtain the prediction of the network for a map showing very high pure noise (Gaussian white noise should be OK) after being preprocessed (normalized) by your approach. (max at 1 approximately and min at 0 approximately)." Please include this experiment, the one introduced is noise+simulated protein not pure noise.

Response: Thank you for your valuable feedback. We apologize for not fully understand your previous requirement regarding obtaining the prediction of the network for a map showing very high pure noise. We have now included the results of this experiment in the revised manuscript (Supplementary Fig. 5e), and discussed them accordingly. The results show that our approach is also robust to the pure noise and produces an output with minimal artifacts. We believe that this experiment provides additional insights into our proposed method and we thank you for bringing it to our attention.

[Page 18, 3rd paragraph; Supplementary Fig. 5e]

6) About the plots of loss vs epochs, Supplementary figure 4 is difficult to understand because it shows the loss vs. the architecture + loss type. So, the Y-axis value depends on the configuration they evaluate. They should add at least two other plots that are L1 vs epochs, SSIM vs epochs for each of the tested architectures. Then we could better compare the effect of loss type, since in your tables (Supplementary Data 5 and 6) it is difficult to appreciate any effect, ~0.05 - 0.07A difference is significant? For me it does not seem to be. In these tables you must report a measure of dispersion, such as the standard deviation. Better than these sup tables would be much better to show boxplots or violinplots.

Response: We appreciate the reviewer's feedback on the presentation of our results. We agree that the figure may not be the most informative for evaluating the effect of loss type on the training process. Therefore, we have added two additional plots to show L1 and SSIM losses versus epochs for each of the tested architectures separately.

[Supplementary Fig. 7a-d]

Following the reviewer's suggestion, we have included the standard deviations of the results in the Supplementary tables regarding the ablation results. We have also included the boxplots in the revised manuscript to provide a clearer visualization of the distribution of the ablation results. In addition, we have conducted the Wilcoxon signed-rank test between each ablation model and the baseline model. It is shown that the differences of unmasked FSC-0.5 and Q-score between the baseline model and the ablation models are statistically significant. We have also moved the table of ablation results from Supplementary Information to the main text of the revised manuscript and discussed them further, so as to highlight the importance of the proposed methods in

achieving superior results.

[Page 19, last paragraph; Page 20, 1st paragraph; Page 39, Table 4; Supplementary Fig. 6; Supplementary Data 7 and 8]

7) Along the revised paper (and in this response to reviewer document) authors have mentioned many times the following idea about the major advancements of EMReady: “the local and non-local modeling capability in the SCUNet network as well as the combination of local smooth L1 loss and non-local SSIM loss in the training process.” In the response to referee 6) Network architecture, they include the following paragraph in the paper: “Despite the powerfulness of non-local modeling in EMReady, it should be noted that the nonlocal range cannot be global but have a limited distance because of the high memory cost of swin transformer during the training of EMReady. Namely, the widows size of swin transformer and the size of input density box, which affect the non-local modeling range of EMReady, is strongly limited by the total memory of GPUs during the training, though the running of the trained EMReady model is not memory expensive and can be accommodated on a GPU with a memory as low as 8G. It is expected that the non-local modeling capability of EMReady can be further improved with more advanced GPUs with higher memory available.”

As I mentioned in my previous comment, “the authors process the map in chunks of 48x48x48 voxels, claiming that they are considering non-local features is a bit misleading” Thus, the non-local capacities of EMReady is restricted to regions of 48x48x48 voxels, which is quite local. Authors should indicate this limitation (with the actual numbers) explicitly in the paper and relax most of the sentences about the capacity of EMReady to capture local and non-local information and about the novelty/major advances of their approach and in practice it is not the case. The authors are free to claim that they use newer architectures and network components, but they shouldn't claim that non-locality is the main reason for the better performance.

Response: Thank you for bringing up this point. We agree that the non-local modeling capability of EMReady is limited to regions of 48x48x48 voxels due to the memory constraints during training. We have made sure to explicitly mention this limitation with actual numbers, and changed the corresponding descriptions to accurately reflect the capabilities of EMReady. Additionally, our claims regarding the novelty and major advances of our approach have been adjusted accordingly.

[Page 4, at the bottom; Page 19, 1st sentence of Section 2.7; Page 20, on the top; Page 22, in the middle of 1st paragraph]

8) Map-model Fourier shell correlation. I appreciate the effort done by authors including the 0.5 FSC resolutions for unmasked maps in supplementary tables. However, as I mentioned before, FSCs are extremely sensitive to used masks. Additionally, reducing the FSC to a single number is a huge lost of information. Thus, I see extremely important to show along the different figures of the paper (not only in supplementary material) always plots of unmasked FSCs. If authors want to show

masked FSCs as well, it is OK, but always the unmasked plots should be shown to avoid misleading conclusions.

Response: We understand your concerns about the presentation of our results and agree that the unmasked FSCs should be included as our main results, instead of masked FSCs. We have updated the manuscript according to your suggestion. We have also made sure that unmasked FSC plots are always shown along different figures in our revised manuscript and that our discussions are based on the unmasked FSCs.

[The entire manuscript]

9) Weird results of DeepEMHancer. First, I see this response a very important example of why including unmasked FSC results in the main manuscript is absolutely mandatory. Masked FSC results are biased by the solvent mask used and could provide misleading results. Please include ALL unmasked results in the manuscript and NOT ONLY in supplementary tables and figures. I do see this point totally mandatory. Additionally, I do not understand the explanation provided by the authors. When you mention ‘deposited primary map’, I (and likely everyone in the field) would think that this is the refined map before any postprocessing step. Additionally, the deposited map FSC resolution is the one provided from half maps without any postprocessing step. But according to your comment, this is an actual postprocessed map and because of this “the input maps are primary maps that usually have been masked/sharpened by their authors, thus are hard for DeepEMhancer to further improve”. To me this does not take any sense. If the primary map deposited in EMDB is postprocessed, then you should take the average from half maps. I continue thinking that authors should clearly explain the discrepancies they obtained between the DeepEMhancer results expected according to the literature and what they obtained in their results and that this should be included in the main text. Second thing is directly asking to authors, if in their test sets they have explicitly looked for deposited maps, which were processed by DeepEMhancer in their publications. In this case, they results may be biased and their conclusions compromised. Can you please provide the percentage of maps in the test set that were processed by DeepEMhancer in their publications?

As these comparisons are your primary results supporting your method, I do see this discussion very worrying. Also, I am concerned about authors providing only masked FSC results in the main text and main figures, and only some raw information in supplementary material without any discussion. The only thing authors said about these unmasked results is: “In addition to the masked map-model FSC, we also report the unmasked map-model FSC-0.5 (Supplementary Table 1), where the significantly better performance of EMReady than other methods can also be observed.”

“We have also published all the input and post-processed maps of DeepEMhancer to our website (<http://huanglab.phys.hust.edu.cn/EMReady/>). [Page 29, Data availability]”

Sorry, I cannot find these results in the website.

Response: We thank the reviewer for raising the concern. We agree with the reviewer that masked FSC results may be biased by the solvent mask used and could provide misleading results. Following the reviewer’s suggestion, we have replaced all the masked FSC results with the unmasked FSCs in the entire manuscript and figures. We hope that these changes can better convey the performance of our method.

[The entire manuscript]

Regarding the confusion about the deposited primary map, we apologize for the lack of clarity in our initial response. To clarify, the primary maps deposited in EMDB are often postprocessed and deposited in their final form, making them challenging for DeepEMhancer to further improve. Our intention was to evaluate whether EMReady can further improve such primary maps and compare it with DeepEMhancer. In addition to the primary maps, we evaluated DeepEMhancer on the test set of half-maps and compared it with our EMReady (Section 2.3). We apologize for not explaining it clearly, and have revised the text to provide a more accurate and detailed description.

[Page 6, 2nd sentence of Section 2.2.1]

In response to your question about the test set, it is confirmed from the Header Metadata in EMDB and through literature search that none of the primary maps in our test set were processed by DeepEMhancer.

We would like to point out that the discrepancies of DeepEMhancer observed by the reviewer were due to the use of masked FSC metric, and not by the performance of DeepEMhancer. As we have shown in our revised manuscript, the unmasked FSCs are significantly improved by DeepEMhancer, which is consistent with the evaluation results in its paper.

[Page 36, Table 1; Page 37, Table 2; Page 41, Fig. 2a,b; Page 44, Fig. 5a,b]

As for the website, thank you for bringing this to our attention. We apologize for any confusion caused by the link provided in our previous response, although a download link for the data was provided on our <http://huanglab.phys.hust.edu.cn/EMReady/> website. The exact link for the input and post-processed maps of DeepEMhancer is <http://huanglab.phys.hust.edu.cn/EMReady/data/>. We have updated the Data Availability section in the manuscript to reflect this.

[Page 32, Data availability]

10) Previous referee comment (15)

“Following the reviewer’s suggestion, we have displayed the raw EMReadyprocessed map at a very low level in Supplementary Fig. 3d. It can be seen from the figure 12 that the noise is not structured. There does not exist hallucination. [Page 17, at the bottom of 1st paragraph; Supplementary Fig. 3d]”

To my eye, Supplementary Fig. 3, show alpha helix like structures in the lipid region. Although the helix-like structure resemblance could be argued, what it is undeniable from this figure is that the noise is always not structured.

Response: Thank you for bringing this to our attention. We agree that we should not claim that the noise is always not structured. We have revised the statement in our manuscript accordingly, as “It can be seen from the figure that the noises introduced by the lipid nanodisc are suppressed and only appear in the EMReady-processed map at very low thresholds. More importantly, one can see that the noises processed by EMReady do not seem to add significant structural artifacts to the density of macromolecule, compared with those in the deposited map.”.

[Page 18, 2nd paragraph]

11) Previous referee comment (16). I do not see the answer providing by authors satisfying. Both map-to-map and map-to-model FSCs should go to zero at Nyquist. This is a typical sanity check. Authors should investigate why FSCs obtained from EMReady processed maps does not go to zero when compared to atomic models.

Response: Thank you for your comment regarding the Nyquist frequency. We agree that both map-to-map and map-to-model FSCs should approach zero at the Nyquist frequency, which defines the resolution limit of 3D reconstructions. We understand that this is a crucial sanity check for cryo-EM reconstructions, and apologize for not providing a satisfactory answer in our previous response. It is known that experimental cryo-EM 3D reconstructions are limited by the Nyquist frequency, as high-frequency information beyond this limit is indistinguishable from noise due to the finite sampling rate during data acquisition, which cannot be retained in the final reconstruction. However, there are cases where high-frequency information beyond the Nyquist limit can be added or enhanced in the map. For example, interpolating a deposited 3D reconstruction to a larger grid size retains high-frequency information and can lead to FSC curves that exceed the Nyquist limit (see the figure below). Similarly, post-processing methods can also add or enhance high-frequency information beyond the Nyquist limit, as shown in the FSC curve of DeepEMhancer in the figure below. In the case of EMReady, the simulated map contains high-frequency information since it is directly determined from the PDB model. Therefore, by training on pairs of experimental cryo-EM maps as input and simulated data as ground truth, EMReady can add or enhance high-frequency information beyond the Nyquist limit, as illustrated in the figure below. We appreciate your feedback and hope that this explanation can clarify any confusion on the topic.

Reviewer #3 (Remarks to the Author):

The authors present a cryo-EM map post-processing approach, called EMReady. It is based on a 3D convolutional neural network that is trained on pairs of experimental cryo-EM maps as input and simulated data as ground truth. They present a series of tests for evaluating the performance of the model, primarily in terms of improved atom resolvability and map interpretability, and results that support improvements in some of these aspects over other methods, like DeepEMhancer. Although the approach is novel and has great potential, the evaluation is lacking and there are several unsubstantiated claims made by the authors.

Response: Thank you for taking the valuable time to review our manuscript and for your positive feedback regarding the novelty and potential of our EMReady approach. We appreciate your constructive criticism and have carefully considered your comments regarding the evaluation and claims made in the manuscript. We have revised the manuscript to address your concerns and provide more comprehensive evaluation of our approach. We hope that the updated version meets your expectations.

MAJOR ISSUE 1: Map interpretability using phenix.map to model

Phenix.map to model is no longer state-of-the-art in automated model building. Additionally, it is reliant on a series of map segmentations that should particularly benefit from data-driven segmentation preprocessing. There are several deep-learning based methods that perform automated model building today. This includes Deept racer,

MAINMAST and ModelAngelo. The authors are strongly encouraged to evaluate the sequence coverage and/or recall using one or two of these methods after processing the map using EMReady. If map quality is truly improved (beyond just improved segmentation or chain separation), this should be reflected in the results from these methods as well.

Response: Following the reviewer's suggestion, we have evaluated the map interpretability using MAINMAST in the revised manuscript and compared EMReady with other methods including DeepEMhancer and phenix.auto_sharpen. Specifically, on the test set of 385 protein chains, EMReady-processed maps yielded significantly improved coverage and sequence match of 85.6% and 33.8%, respectively, compared to 73.9% and 15.2% for the deposited map, 72.1% and 14.6% for DeepEMhancer-processed map, and 74.0% and 15.4% for phenix.auto_sharpen-processed maps. The evaluation results suggest the truly improvement of map interpretability by EMReady. [Page 13, last paragraph; Page 30, 1st paragraph; Page 38, Table 3; Page 47, Fig. 8; Supplementary Data 6]

MAJOR ISSUE 2: Atom resolvability using Q-score and model-to-map FSC

The Q-score measures how well the experimental maps fit with Gaussian-like functions around identified atoms. The authors present a neural network that is trained on simulated ground truth data that is generated using a Gaussian forward model, see equation (1). Hence, the L1-loss used in the training of the model is indirectly also maximizing the Q-score. Naturally, the output of any processing using this network will exhibit more Gaussian-like densities, which will inflate the Q-score. Subsequently, there is little to no value in using the Q-score to evaluate the performance of the method as this particular metric was indirectly optimized throughout training. The same applies also to model-to-map FSC, as the model is first converted to a gaussian mixture model before the FSC is calculated.

Response: We thank the reviewer for the comment on the use of the Q-score and model-to-map FSC in evaluating the performance of EMReady. We agree that the Q-score and model-to-map FSC are indirectly optimized during the training of EMReady as it is based on a Gaussian forward model. We have explicitly pointed out the limitations of using Q-score and model-to-map FSC as the measurement of performance in our revised manuscript. However, we still believe that the Q-score and model-to-map FSC are useful metrics for evaluating the quality of the maps processed by EMReady on the test set. On one hand, since we evaluate the performance of EMReady on an independent test set relative to the training set, Q-score and model-to-map FSC are valuable to estimate the model's performance on independent data and detect whether EMReady overfits or underfits to the training data. On the other hand, these metrics are basic and widely used measurements of the fit between the experimental maps and the atomic structures. Besides, it is important to note that we also present other metrics including the CC values, conduct other experiments like calculating the FSC between processed and unprocessed half-maps and the FSC against higher-resolution maps, and

provide the visual inspection of the maps, to provide a comprehensive evaluation of the performance of EMReady. We hope that these can address the reviewer's concern regarding the use of the Q-score and model-to-map FSC.

[Page 7, 2nd paragraph]

MAJOR ISSUE 3: Map quality

Although the map becomes perhaps more interpretable to humans there are no results indicating improvements to the quality of the reconstruction itself through for instance denoising or local sharpening/filtering. The generated maps increase similarity to a simple gaussian forward model, which might not be the true underlying signal that is observed in cryo-EM. This is however neither evaluated or clarified by the authors. The authors could evaluate this by for instance applying EMReady to one of the half-maps and calculate the FSC between the output and the other unprocessed half-map. Notice that the FSC_{0.5} between two processed half-maps will be inflated due to injection of the same signal/bias into both half-maps, which is similar to the solvent mask effect, see (Chen Ultramicroscopy 2013).

Response: We very appreciate the reviewer's suggestion to calculate the FSC between one processed half-map and the other unprocessed half-map as a means of evaluating the effectiveness of EMReady in preserving the underlying signal observed in cryo-EM. We have conducted additional experiments using this approach. Our evaluation results showed that the FSC-0.5 values between the processed and unprocessed half-maps were indeed improved, which proved that EMReady did capture the true underlying signal that is observed in cryo-EM and improved the quality of the reconstruction itself.

[Page 11, last paragraph; Page 12, 1st paragraph; Page 46, Fig. 7; Supplementary Data 4]

MAJOR ISSUE 4: Method applicability and overall evaluation

Although the authors mention at the end of the discussion section that “users should not solely rely on the EMReady-processed map when building a model from a cryo-EM map”, more results need to be presented that evaluate when the approach is expected to fail. For instance, what are the outliers in Figure 7a and what actually happens to ligands or lipids?

Response: We agree that it is important to evaluate the limitations of our approach. We have addressed this issue in our revised manuscript. Specifically, we have discussed the cases where our EMReady approach is likely to fail, and provided the corresponding examples/figures for illustration. We hope these additional results and discussions provide a better understanding of the limitations of our approach and help users take more advantages of our EMReady method.

[Page 23, last paragraph; Supplementary Fig. 8]

MINOR ISSUES

Why is the validation and testing dataset so large relative to the training dataset?

Response: We appreciate the reviewer's comment on the size of our validation and testing datasets compared to the training dataset. As described in Section 4.2, we collected a total of 436 high-quality maps with CC_mask values of above 0.75. Out of these cases, we randomly selected 86 for the test set, 280 for the training set, and used the remaining 70 maps as the validation set. Namely, about 20% cases are separated from the total of 436 cases as the test set. From the rest 350 maps, 20% cases are selected as the validation set. Additionally, we added 24 lower-quality maps with CC_mask values below 0.75 to the test set. Our reason for having a relatively larger validation and testing dataset is to ensure the generalizability of the model to unseen data. By having a larger validation and testing dataset, we can better estimate the model's performance on unseen data and detect any overfitting or underfitting problems. The size of our testing and validation sets is consistent with what is commonly used in the realm of machine learning and deep learning. We hope that this explanation addresses the reviewer's concern and provides insight into our methodology.

Why are the volume chunks “48x48x48”? How does the performance change with chunk-size? The batch-size is unusually large. Have the authors tried smaller batch-sizes? How does this affect convergence and overfitting? What about dropout and weight decay?

Response: We thank the reviewer for raising the questions. The volume chunks are set to 48x48x48 to balance the memory cost and the ability to capture local information. We have already experimented with smaller chunk sizes and found that the performance does decline with smaller chunk sizes. These results are summarized in Table 4 and detailed in Supplementary Data 7 and 8.

We have experimented with a smaller batch size of 27, and found that the computation time required to ensure convergence is drastically increased but the performance is almost the same. Therefore, we chose a larger batch size of 108, mainly for computational efficiency. We have also experimented with dropout and weight decay, and found that the performance slightly decreased with these regularization techniques. Therefore, we did not use them in our final model. We have added the learning curves and evaluation results in our revised manuscript and discussed them accordingly.

[Page 28, in the middle of 1st paragraph; Supplementary Fig. 9; Supplementary Data 12-14]

Overall, we have carefully considered various hyperparameters and experimented with different settings to optimize the performance of our method. The final choices were based on empirical observations and computational efficiency.

The word “chunk” is used extensively throughout the results and method section to refer to volume slices. The word slice, patch or section are more suited alternatives.

Response: Following the reviewer's suggestion, we have changed the word "chunk" to "slice" in the revised manuscript.

[The entire manuscript]

The use of the word "primary map" is used to refer to final reconstruction results as opposed to half-maps. This needs to be explicitly clarified as this phrase lacks a general meaning within the field.

Response: We thank the reviewer for pointing out the issue. We have revised the text to explicitly clarify that the "primary map" refers to the final reconstruction results, as opposed to half-maps.

[Page 6, 2nd sentence of Section 2.2.1]

Reviewers' Comments:

Reviewer #2:

Remarks to the Author:

In this revision, the authors have significantly improved the article. I appreciate the work done by the authors. I have a few additional comments that I believe will not take too much time/effort for them:

1) I appreciate the inclusion of Suppl Fig. 1. However, in my opinion, this figure does not show a fair comparison between the different postprocessed maps. The thresholds should be carefully selected so that the covered area by the maps is approximately similar. In my view this is not fulfilled and I see the results clearly biased.

2) In the new figure Suppl Fig. 5e (especially for the 0.05 and 0.1 thresholds) I can see again a clear structural noise after EMReady (I again see alpha-like structures), which to me is to be expected and not a very bad thing. However, the authors should indicate the value of this structured noise relative to the value of the original noise map (I guess should be low) and indicate this issue in the text to warn potential users that low density values are risky.

3) It is important to run EMReady on raw maps without post-processing and on post-processed maps to see in which cases it works best.

Reviewer #3:

Remarks to the Author:

After carefully reviewing the revised manuscript, I believe the authors have sufficiently addressed the major issues. Therefore, I recommend that the manuscript be published at this time.

Manuscript ID: NCOMMS-22-31369B

Title: Improvement of cryo-EM maps by simultaneous local and non-local deep learning

Author(s): Jiahua He; et al.

We very much appreciate the valuable comments/suggestions from the two reviewers. We have conducted necessary computations/analyses and revised our manuscript accordingly. The revised parts in the manuscript are highlighted **in red**. The point-to-point responses to the comments are listed as follows.

Reviewer #2 (Remarks to the Author):

In this revision, the authors have significantly improved the article. I appreciate the work done by the authors. I have a few additional comments that I believe will not take too much time/effort for them:

Response: Thank you very much for your positive feedback and constructive comments on our revised manuscript. We greatly appreciate your time and effort in providing us with valuable feedback to help enhance the quality of our work. We have carefully considered your additional comments and made necessary revisions to further improve our article. We hope that the revised manuscript addresses your concerns and meets your expectations.

1) I appreciate the inclusion of Suppl Fig. 1. However, in my opinion, this figure does not show a fair comparison between the different postprocessed maps. The thresholds should be carefully selected so that the covered area by the maps is approximately similar. In my view this is not fulfilled and I see the results clearly biased.

Response: We greatly appreciate your feedback and apologize for any bias caused by the figure. Follow your suggestions, to ensure a fair comparison, we have carefully selected appropriate contour thresholds to ensure that the covered volumes (measured by Chimera) by different maps of each case in the figure are approximately equal.

[Supplementary Fig. 1]

2) In the new figure Suppl Fig. 5e (especially for the 0.05 and 0.1 thresholds) I can see again a clear structural noise after EMReady (I again see alpha-like structures), which to me is to be expected and not a very bad thing. However, the authors should indicate the value of this structured noise relative to the value of the original noise map (I guess should be low) and indicate this issue in the text to warn potential users that low density values are risky.

Response: Thank you for raising this important issue. Following your suggestion, we

have thoroughly discussed the issue of structural noise and indicated the relative value in the revised manuscript, as “Although we notice that the agglomeration of noise may lead to some structural patterns (Supplementary Fig. 5e), the density values for structured noises are extremely low in the EMReady-processed map compared to those for the underlying macromolecule (Supplementary Fig. 5a,d). For example, structured noises can be seen at density thresholds of 0.05 and 0.1, which approximately correspond to density values of respectively 1/6 and 1/3 in the original noise map (calculated by re-scaling the EMReady-processed map to have an equal standard deviation)”. We have also indicated this issue in the revised manuscript as “it should be emphasized that low density values are risky and should be carefully handled by potential users when interpreting the EMReady-processed map”.

[Page 18, at the bottom; Page 19, on the top; Supplementary Fig. 5a]

3) It is important to run EMReady on raw maps without post-processing and on post-processed maps to see in which cases it works best.

Response: Thank you for the valuable comments. Follow your suggestion, we have compared the evaluation results of EMReady using unprocessed half-maps and using post-processed primary map as the input on the test set of 25 pairs of half-maps. Our results reveal that EMReady performs better on post-processed primary maps than on unprocessed half-maps. These results can be understood because EMReady is trained on post-processed primary maps. It is expected that EMReady can be further improved through training on unprocessed half-maps when more and more half-maps become available for their PDB structures. We have added a separate paragraph to discuss about this issue.

[Page 23, 3rd paragraph; Supplementary Fig. 8]

Reviewer #3 (Remarks to the Author):

After carefully reviewing the revised manuscript, I believe the authors have sufficiently addressed the major issues. Therefore, I recommend that the manuscript be published at this time.

Response: Thank you very much for your thorough review of our revised manuscript and for recommending its publication. We greatly appreciate your time and effort in providing us with constructive feedback to help improve the quality of our work.